# WATCH: Adaptive Monitoring for AI Deployments via Weighted-Conformal Martingales

Drew Prinster [1]   Xing Han [1]   Anqi Liu [1]   Suchi Saria [1]

## Abstract

Responsibly deploying artificial intelligence (AI) / machine learning (ML) systems in high-stakes settings arguably requires not only proof of system reliability, but also continual, post-deployment monitoring to quickly detect and address any unsafe behavior. Methods for nonparametric sequential testing—especially conformal test martingales (CTMs) and anytime-valid inference—offer promising tools for this monitoring task. However, existing approaches are restricted to monitoring limited hypothesis classes or "alarm criteria" (e.g., detecting data shifts that violate certain exchangeability or IID assumptions), do not allow for online adaptation in response to shifts, and/or cannot diagnose the cause of degradation or alarm. In this paper, we address these limitations by proposing a weighted generalization of conformal test martingales (WCTMs), which lay a theoretical foundation for online monitoring for any unexpected changepoints in the data distribution while controlling false-alarms. For practical applications, we propose specific WCTM algorithms that adapt online to mild covariate shifts (in the marginal input distribution), quickly detect harmful shifts, and diagnose those harmful shifts as concept shifts (in the conditional label distribution) or extreme (out-of-support) covariate shifts that cannot be easily adapted to. On real-world datasets, we demonstrate improved performance relative to state-of-the-art baselines.

## 1. Introduction

As AI/ML systems become integral to real-world applications, ensuring their safety and utility under evolving conditions is essential for responsible deployment. However, even meticulously trained models with apparent reliability guarantees can fail abruptly when shifts in the data distribution or operational environment violate the underlying conditions they were designed for (Amodei et al., 2016). That is, unforeseen deployment conditions can make it impossible to guarantee reliability for all situations in advance. Consequently, there is growing recognition for the need to continuously monitor deployed AI systems for determining when model updates are required to mitigate downstream harm (Vovk et al., 2021; Podkopaev and Ramdas, 2021b; Feng et al., 2022; 2025). In this work, we moreover argue that such monitoring methods should ideally perform at least three key functions: (1) maintain end-user reliability and minimize unnecessary alarms by adapting online to mild or benign data shifts; (2) rapidly detect more extreme or harmful shifts that necessitate updates; and (3) identify the root-cause of degradation to inform appropriate recovery.

For example, consider a healthcare use case where the goal is to predict the risk of sepsis, a life-threatening infection, $Y$ from electronic health record inputs $X$ (e.g., vital signs, lab tests, medical history, demographics) using an AI/ML system (e.g., Adams et al. (2022)). Various clinical data shifts pose challenges to monitoring in practice (Finlayson et al., 2021). Figure 1 illustrates hypothetical synthetic-data shifts that can be interpreted through this sepsis example, where for simplicity $X$ only represents a patient's age. An example of a benign shift is a mild shift in patient demographics primarily toward young adults; Figure 1a shows a corresponding covariate shift in the marginal $X$ distribution. Such a mild shift to a younger population would be benign, as younger individuals are well-represented in the training data and also tend to have lower, less variable (easier to predict) sepsis risk—so, there is no need for an alarm. On the other hand, harmful shift examples include if the AI tool were deployed with a much older population than observed in the training data (e.g., Figure 1b) or if a new microbial strain were to arise that was especially severe in children (e.g., Figure 1c). In any such harmful shift case, swift detection and root-cause analysis is essential to initiate and inform retraining, to ultimately minimize harm.

Recent advances in anytime-valid inference (Ramdas et al.,

[1]Department of Computer Science, Johns Hopkins University, Baltimore, MD, USA. Correspondence to: Drew Prinster <drewprinster@gmail.com>, Xing Han <aaronhan223@gmail.com>.

*Proceedings of the 42nd International Conference on Machine Learning*, Vancouver, Canada. PMLR 267, 2025. Copyright 2025 by the author(s).

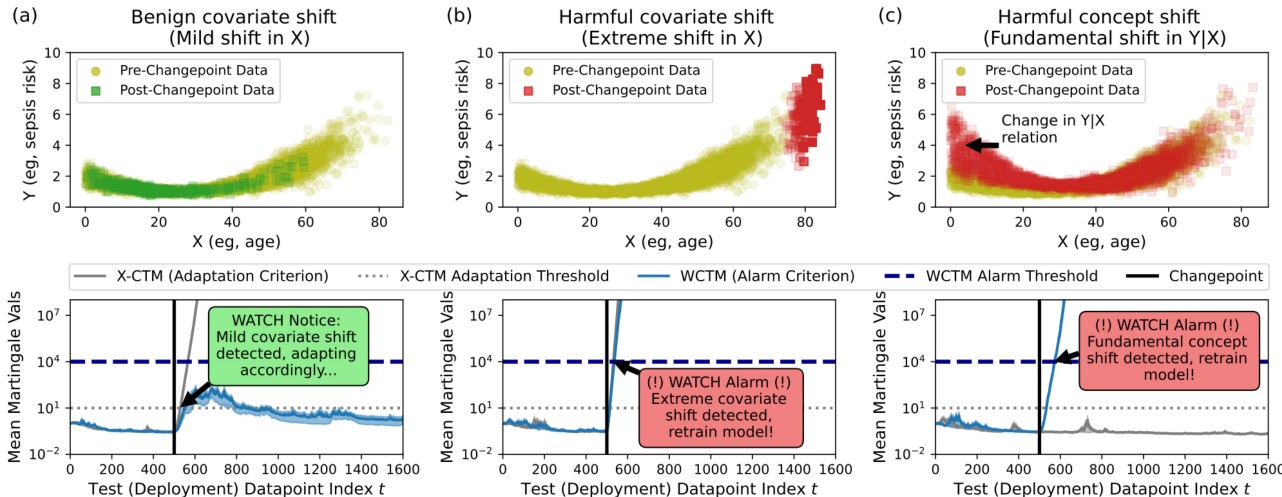

Figure 1: Each column represents a data shift scenario: the top row is a simulated shift example and the bottom row shows WATCH's response, averaged over 20 random seeds. WATCH raises an alarm to retrain the AI/ML once the WCTM (blue) exceeds its alarm threshold; meanwhile, an $X$-CTM (gray)—a standard CTM that only depends on inputs $X$, and thus only detects covariate shifts—dynamically initiates the WCTM's adaptation phase and aids in root-cause analysis. In (a), the $X$-CTM starts the WCTM's adaptation phase, which allows the WCTM to avoid raising an unnecessary alarm. In (b), the extreme covariate shift causes the WCTM to raise an alarm, indicating that the covariate shift is too severe to be adapted to. In (c), the illustrated concept shift causes WATCH to raise an alarm, but without the $X$-CTM detecting a shift in covariates $X$—this allows WATCH to diagnose the root-cause of the alarm as a concept shift in $Y \mid X$.

2023) and especially conformal test martingales (CTMs) (Volkhonskiy et al., 2017; Vovk, 2021) offer promising tools for AI monitoring with sequential, nonparametric guarantees. However, existing CTM monitoring methods (e.g., Vovk et al. (2021)) all rely on some form of *exchangeability* (e.g., IID) assumption in their null hypotheses—informally, meaning that the data distribution is the same across time or data batches—and as a result, standard CTMs can raise unnecessary alarms even when a shift is mild or benign (e.g., Figure 1a). Meanwhile, existing comparable monitoring methods for directly tracking the risk of a deployed AI (e.g., Podkopaev and Ramdas (2021b)) tend to be less efficient than CTMs in their speed of detecting harmful shifts, data usage, and/or computational complexity (Sec. 4.3).

Our paper's contributions can be summarized as follows:

- Our main theoretical contribution is to propose weighted-conformal test martingales (WCTMs), constructed from sequences of online weighted-conformal $p$-values, which generalize their standard conformal precursors. WCTMs lay a theoretical foundation for sequential and continual testing of a broad class of null hypotheses beyond exchangeability, such as shifts that one aims to model and adapt to.

- For practical applications, we propose WATCH: **W**eighted **A**daptive **T**esting for **C**hangepoint **H**ypotheses, a framework for AI monitoring using WCTMs. WATCH continuously adapts to mild or benign distribution shifts (e.g., Figure 1a) to maintain end-user safety and utility (and avoid unnecessary

alarms), while quickly detecting harmful shifts (e.g., Figure 1b & c) and enabling root-cause analysis.

## 2. Background

**Notation:** Assume an initial dataset $Z_{1:n} := \{Z_i\}_{i=1}^n$, where each datapoint is a feature-label pair, $Z_i := (X_i, Y_i) \in \mathcal{X} \times \mathcal{Y} = \mathcal{Z}$. For simplicity, further assume that an AI/ML model $\widehat{\mu}$ is pretrained on a separate dataset.[1] After deploying the AI/ML model, the test points are observed sequentially at each time $t = 1, ..., T$ (batch data can be given random ordering), though for simpler exposition, we initially focus on $t = 1$. We abbreviate indices $[m] := \{1, ..., m\}$. Random variables are denoted with capital letters (e.g., $Z_i$) and observed values with lowercase (e.g., $z_i$). For $Z := Z_{1:(n+t)}$, we let $F_Z$ denote the joint distribution function, $f_Z$ the joint density function. For a set of distributions $\mathbf{F_Z}$, we write $Z_1, Z_2, ... \sim \mathbf{F_Z}$ to mean the sequence has some unknown distribution $F_Z \in \mathbf{F_Z}$.

### 2.1. Conformal Prediction and Conformal $p$-Values

Standard conformal prediction (CP) (Vovk et al., 2022) is an approach to *predictive inference*: the task of converting a black-box AI/ML prediction, $\widehat{\mu}(X_{n+1})$, into a *predictive confidence interval/set*, $\widehat{C}_{[n],\alpha}(X_{n+1})$, that should contain the true label with a user-specified rate, $1 - \alpha \in (0, 1)$ (e.g.,

---

[1]This corresponds to a split conformal setting (Papadopoulos, 2008), but our theory also extends to full conformal (Vovk et al., 2022), which avoids splitting data at a heavy computational cost.

90%). This objective is called valid (marginal) *coverage*:[2]

$$\mathbb{P}\{Y_{n+1} \in \widehat{C}_{[n],\alpha}(X_{n+1})\} \geq 1 - \alpha. \quad (1)$$

Constructing the CP set, $\widehat{C}_{[n],\alpha}(X_{n+1})$, requires a labeled calibration dataset,[3] $Z_{1:n}$, and a "nonconformity"*score function*, $\widehat{\mathcal{S}} : \mathcal{X} \times \mathcal{Y} \to \mathbb{R}$, which generally uses the prefit ML predictor to quantify how "strange" a point $(x, y)$ is relative to the training data. A common example is the absolute-value residual score, $\widehat{\mathcal{S}}(x, y) = |y - \widehat{\mu}(x)|$.

Though not always framed as such, the CP set $\widehat{C}_{[n],\alpha}(X_{n+1})$ can be computed via a *conformal p-value*, which places the "strangeness" of the test point's score in context of the calibration-data scores. That is, with $v_i := \widehat{\mathcal{S}}(x_i, y_i)$ for all $i \in [n + 1]$, Vovk et al. (2022) defines the conformal $p$-value for test point $Z_{n+1}$ relative to calibration data $Z_{1:n}$ as the fraction of the $n + 1$ scores, $v_{1:(n+1)}$, that are at least as large as $v_{n+1}$, with ties broken uniformly at random:

$$p_{n+1} := \frac{|\{i : v_i > v_{n+1}\}| + u_{n+1}|\{i : v_i = v_{n+1}\}|}{n + 1}, \quad (2)$$

where $U_{n+1} \stackrel{iid}{\sim} \text{Unif}[0, 1]$; that is, $u_{n+1}$ is obtained from an independent standard uniform distribution.[4] The CP set $\widehat{C}_{[n],\alpha}(X_{n+1})$ can then be understood as the subset of labels $y \in \mathcal{Y}$ that would not result in too "extreme" of a test-point $p$-value, $p_{n+1}(X_{n+1}, y)$ (this is explicit notation for $p_{n+1}$ to emphasize its dependence on the candidate label $y$):

$$\widehat{C}_{[n],\alpha}(X_{n+1}) := \{y \in \mathcal{Y} : p_{n+1}(X_{n+1}, y) > \alpha\}. \quad (3)$$

## 2.2. Exchangeability Underlies Standard CP Validity

For the standard CP set in Eq. (3), the coverage guarantee in Eq. (1) holds assuming that the calibration data $Z_{1:n}$ and test point $Z_{n+1}$ are all *exchangeable*—intuitively, this means that the data distribution is invariant over time (e.g., independent and identically distributed or "IID" data). Formally, exchangeability means that the joint distribution, $F_Z$, is invariant to reorderings or permutations $\sigma$: that is, $F_Z(z_{\sigma(1)}, ..., z_{\sigma(n+1)}) = F_Z(z_1, ..., z_{n+1})$ for all $z_1, ..., z_{n+1} \in \mathcal{Z}$ and all $\sigma : [n + 1] \to [n + 1]$. With $\mathbf{F}_{\mathbf{Z}}^{\mathbf{ex}}$ denoting the set of exchangeable joint distributions, we can write the assumption of exchangeability as a (composite and nonparametric) null hypothesis:

$$\mathcal{H}_0^{\text{ex}} : Z_1, Z_2, ..., Z_{n+1} \sim \mathbf{F}_{\mathbf{Z}}^{\mathbf{ex}}. \quad (4)$$

---

[2]Here, marginal means on average over the draw of calibration and test data; see, e.g., Foygel Barber et al. (2021) for more details.

[3]For $t > 1$, the calibration data might be kept fixed (i.e., $Z_{1:n}$), or it may include past test points (i.e., $Z_{1:(n+t-1)}$); for now, we focus on $t = 1$ to avoid this distinction and simplify exposition.

[4]Conservative conformal $p$-values set $u_{n+1} := 1$; the random variable $P_{n+1}$ corresponding to $p_{n+1}$ is called a $p$-variable, but we will often refer to both as $p$-values for more common terminology.

Standard conformal $p$-values (Eq. (2)) are valid or "bonafide" $p$-values, in the usual statistical testing sense, for the exchangeablity null hypothesis (Vovk et al., 2022). That is, assuming $\mathcal{H}_0^{\text{ex}}$, then $P_{n+1} \stackrel{iid}{\sim} \text{Unif}[0, 1]$ (Vovk, 2021) and

$$\mathbb{P}_{\mathcal{H}_0^{\text{ex}}}(P_{n+1} \leq \alpha) \leq \alpha \qquad \forall \alpha \in (0, 1). \quad (5)$$

Thus, observing an extreme value of $p_{n+1}$ is evidence against $\mathcal{H}_0^{\text{ex}}$ (Eq. (4)), or evidence for a distribution shift.

## 2.3. Standard Conformal Test Martingales: Testing Exchangeability Online via Betting

Standard conformal test martingales (CTMs) (Volkhonskiy et al., 2017; Vovk, 2021; Vovk et al., 2021) continually aggregate information from a sequence of conformal $p$-values (Eq. (2)) to perform *online* testing of the exchangeability null, $\mathcal{H}_0^{\text{ex}}$. They can be constructed on top of any standard CP method, and thus on top of any conformalized AI/ML model, and used to monitor for deviations from $\mathcal{H}_0^{\text{ex}}$. We will describe CTMs from a game-theoretic "testing-by-betting" interpretation (Shafer and Vovk, 2019; Shafer, 2021).

Under $\mathcal{H}_0^{\text{ex}}$, a stochastic process $M_0, M_1, ..., M_t, ...$ is a CTM if it is a nonnegative martingale—i.e., $M_t \geq 0$ for all $t$ and $\mathbb{E}_{\mathcal{H}_0^{\text{ex}}}[M_t \mid M_0, ..., M_{t-1}] = M_{t-1}$—constructed from a corresponding sequence of conformal $p$-values $p_1, ..., p_t, ...$ via an appropriately defined "betting process." In the game-theoretic interpretation, a bettor has initial wealth $M_0$ (usually set to $M_0 = 1$, for simplicity). Then, at each time $t = 0, 1, ...$, she may use wealth $M_t$ to bet on the value of $p_{t+1}$ to be observed next; once $p_{t+1}$ is revealed, she receives her reward and/or pays her losses, resulting in a new total wealth $M_{t+1}$. The bettor bets *against* the null hypothesis $\mathcal{H}_0^{\text{ex}}$: if $\mathcal{H}_0^{\text{ex}}$ is true, then $P_t \stackrel{iid}{\sim} \text{Unif}[0, 1]$ and she cannot expect to outperform random guessing; so, if the bettor grows her wealth by a large factor $M_t/M_0 >> 1$, this is evidence that the bettor "knows" an alternative hypothesis more accurate than $\mathcal{H}_0^{\text{ex}}$ (Vovk, 2021). $M_t/M_0$ can be taken as an "anytime-valid" evidence against $\mathcal{H}_0^{\text{ex}}$. In particular, $M_t/M_0$ is also an "$e$-process" for $\mathcal{H}_0^{\text{ex}}$ (see Vovk and Wang (2021); Ramdas et al. (2023)).

More formally, define a *betting function* as a function $h : [0, 1] \to [0, \infty]$ that integrates to one, i.e., $\int_0^1 h(u)\mathrm{d}u = 1$. We follow Vovk (2021) and assume the betting function

$$h_\epsilon(p) := 1 + \epsilon(p - 0.5), \quad (6)$$

where $\epsilon \in \mathbf{E} := \{-1, 0, 1\}$ ($\mathbf{E}$ is selected somewhat arbitrarily). The intuition is that $\epsilon < 0$ corresponds to betting on smaller $p$-values, $\epsilon > 0$ corresponds to betting on larger $p$-values, and $\epsilon = 0$ represents not betting. A CTM can be constructed by selecting a betting strategy $h_{\epsilon_t}$ at each time $t$ that may depend only on the past $p$-value observations $p_1, ..., p_{t-1}$; this paper uses the "composite jumper martin-

gale" strategy described in Vovk et al. (2022). Then, a conformal (test) martingale $M_t : [0,1]^* \to [0,\infty]$, $t = 0, 1, ...,$ is constructed by accumulating the "wealth" attained by the bets on the previous conformal $p$-values, $p_1, p_2, ..., p_{t-1}$. That is, for a single bet $\epsilon_i \in \mathbf{E}$ at each time $i$, the product $\prod_{i=1}^{t} h_{\epsilon_i}(p_i)$ is a CTM that can be interpreted as the factor by which the bettor has multiplied her wealthy by time $t$. More broadly, CTMs may accommodate more flexible betting strategies (while only depending on past $p$-values), $\nu$, that may distribute bets across $\mathbf{E}$ at each time, as

$$M_t := \int \Big( \prod_{i=1}^{t} h_{\epsilon_i}(p_i) \Big) \nu\big(\mathrm{d}(\epsilon_0, \epsilon_1, ...)\big), \ \forall \ t. \quad (7)$$

For example, in our implementations we take $\nu$ to be the composite jumper Markov chain described by Vovk et al. (2022) to determine how the "wealth" values are spread across the betting space $\mathbf{E}$ at each timepoint.

A standard CTM (Eq. (7)) can be used to test the exchangeability null $\mathcal{H}_0^{\mathrm{ex}}$ (Eq. (4)) online either by using $M_t/M_0$ as an anytime-valid evidence metric, or by raising an alarm when $M_t/M_0$ exceeds some user-defined threshold $c$. That is, by Ville's inequality for martingales (Ville, 1939), CTMs achieve anytime-valid control over the false alarm rate (i.e., over the probability of an alarm despite $\mathcal{H}_0^{\mathrm{ex}}$ being true),

$$\mathbb{P}_{\mathcal{H}_0^{\mathrm{ex}}}\big(\exists \ t : M_t/M_0 \geq c\big) \leq 1/c, \quad (8)$$

which is sometimes referred to as *strong validity* due to its control over *ever* raising a false alarm (Vovk, 2021).

### 2.4. Weighted Conformal Prediction for Adapting to Distribution Shifts

Whereas Standard CP computes valid predictive confidence sets assuming exchangeable data, *weighted* conformal prediction (WCP) (e.g., Tibshirani et al. (2019); Podkopaev and Ramdas (2021a); Barber et al. (2023); Prinster et al. (2024); Barber and Tibshirani (2025)) generalizes standard CP to attain valid coverage (Eq. (1)) even under various distribution shifts. WCP methods are thus an approach to *adapting* to distribution shift, that is by computing CP sets on a reweighted version of the empirical calibration set's scores, which in effect can modulate the size of the prediction sets to maintain coverage.

As we will see in the next section, WCP methods are associated with weighted-conformal $p$-values, a special-case of which was introduced in Jin and Candès (2023) for standard covariate shifts, based on Tibshirani et al. (2019). Concurrently with our paper, Barber and Tibshirani (2025) also leverage weighted-conformal $p$-values to unify various theories of conformal prediction. However, there are several key differences: regarding motivation, that work discusses connections to (one-shot) hypothesis testing as a means of

framing conformal prediction, while in our paper (sequential) hypothesis testing is the primary focus; the conformal $p$-values in that paper are deterministic and attain conservative validity, whereas we incorporate randomness to achieve exact validity; lastly, whereas that paper focuses on a batch-data setting where all the data are observed simultaneously, we focus on an online monitoring setting where each $p$-value is determined only by the current and past data, and we importantly present new theory for when a sequence of WCP $p$-values can be distributed uniformly and *independently*, which is important for monitoring with WCTMs.

## 3. Theory and Methods: Weighted-Conformal $p$-Values and Test Martingales

Our main theory and method contribution is to introduce weighted-conformal test martingales (WCTMs), constructed from a generalized version of weighted-conformal $p$-values, which expand the scope of their standard conformal analogs. WCTMs enable more customizable and informative alarm criteria than standard CTMs, and our specific variants enable adaptation to mild shifts, while in response to severe shifts they raise alarms and enable root-cause analysis.

### 3.1. Generalized Weighted-Conformal $p$-Values

In this section we present a general version of weighted-conformal $p$-values.[5] Sequences of these generalized weighted-conformal $p$-values will lay a theoretical foundation for online testing of a broad range of null hypotheses beyond exchangeability. To begin, observe that standard conformal $p$-values (Eq. (2)) can be equivalently written as

$$p_{n+1} = \sum_{i=1}^{n+1} \frac{1}{n+1} \Big[ \mathbb{1}\{v_i > v_{n+1}\} + u_{n+1} \mathbb{1}\{v_i = v_{n+1}\} \Big],$$

where the purpose of this notation is to isolate where the comparison of $v_{n+1}$ to each $i$-th score is given uniform weight $\frac{1}{n+1}$. For some arbitrary weight vector $\tilde{w} = (\tilde{w}_1, ..., \tilde{w}_{n+1}) \in [0,1]^{n+1}$ where $\sum_{i=1}^{n+1} \tilde{w}_i = 1$, it is then straightforward to define *weighted-conformal $p$-values* as

$$p_{n+1}^{\tilde{w}} = \sum_{i=1}^{n+1} \tilde{w}_i \Big[ \mathbb{1}\{v_i > v_{n+1}\} + u_{n+1} \mathbb{1}\{v_i = v_{n+1}\} \Big]. \quad (9)$$

Note that existing (split or full) weighted CP methods (e.g., Tibshirani et al. (2019); Podkopaev and Ramdas (2021a); Prinster et al. (2024) and certain methods in Barber et al. (2023)) can be also be defined by weighted-conformal $p$-values by plugging $p_{n+1}^{\tilde{w}}(X_{n+1}, y)$ in for $p_{n+1}(X_{n+1}, y)$ in

---

[5]Here, "weighted" refers to weights on the score distribution prior to computing a $p$-value, not to weighting the $p$-value itself.

Eq. (3), where $\tilde{w}$ is an appropriately chosen weight vector.[6]

To understand the meaning of the weights $\tilde{w}$ for our hypothesis testing purpose, we draw from the general view of weighted CP described in Prinster et al. (2024), which expounded analysis from Tibshirani et al. (2019): For setup, let $E_z$ denote the event $\{Z_1, ..., Z_{n+1}\} = \{z_1, ..., z_{n+1}\}$, meaning that the empirical distribution of the datapoints has been observed, but we do *not* know whether $Z_i = z_i$, and so on. Then, the oracle weights $\tilde{w}^o$ would be given by entries

$$\tilde{w}_i^o = \mathbb{P}\{V_{n+1} = v_i \mid E_z\} \quad (10)$$
$$= \frac{\sum_{\sigma:\sigma(n+1)=i} f(z_{\sigma(1)}, ..., z_{\sigma(n+1)})}{\sum_\sigma f(z_{\sigma(1)}, ..., z_{\sigma(n+1)})},$$

where $f$ is the joint probability density function (PDF) and $\sigma$ is a permutation of $[n+1]$. In words, the oracle weight $\tilde{w}_i^o$ we would ideally use for $\tilde{w}_i$ is the probability that the test score $V_{n+1}$ took on the value $v_i$, conditioned on the empirical distribution $E_z$. For further exposition, see Prinster et al. (2024). While for arbitrary $f$, computing Eq. (10) is intractable due to requiring knowledge of $f$ and the factorial complexity, we next turn to how simplifying or approximating $\tilde{w}^o$ can be useful in practical hypothesis testing.

## 3.2. Expanded Hypothesis Testing with Weighted-Conformal $p$-Values

We now look at how the weighted-conformal $p$-values introduced in Eq. (9) can be used to sequentially test a variety of nonparametric null hypotheses beyond exchangeability. Let us begin by using $\mathcal{H}_0(\hat{f})$ to denote any set of assumptions on the true $f$ or some approximations on $f$ that we wish to assume are true in our null hypothesis (e.g., which may enable testing for evidence of inaccurate estimation). Hereon let $\tilde{w} := \tilde{w}(\mathcal{H}_0(\hat{f}))$ denote the weight vector computed with this approximation. For example, if we assume exchangeability as our null hypothesis ($\mathcal{H}_0(\hat{f}) = \mathcal{H}_0^{(\text{ex})}$), then $f(z_{\sigma(1)}, ..., z_{\sigma(n+1)}) = f(z_1, ..., z_{n+1})$ and Eq. (10) reduces to $\tilde{w}_i = \frac{1}{n+1}$, which recovers standard conformal $p$-values. More generally, $\mathcal{H}_0(\hat{f})$ may denote some other set of invariance assumptions on $f$ or density-ratio estimates assumed to be accurate (see Sec. 3.3 and App. C for a worked example for our main testing objective). Then, $P_{n+1}^{\tilde{w}}$ is a valid and exact $p$-value for the null hypothesis $\mathcal{H}_0(\hat{f})$:

$$\mathbb{P}_{\mathcal{H}_0(\hat{f})}\{P_{n+1}^{\tilde{w}} \leq \alpha\} = \alpha. \quad (11)$$

Moreover, when a sequence of weighted-conformal $p$-values is computed online, then they are not only distributed uniformly, but also *independently*, at each timestep. We next state this formally.

Theorem 3.1. (*Independence and exact validity, for $\mathcal{H}_0(\hat{f})$, of online WCP $p$-values.*) *For any $T \in \mathbb{N}$, let $\mathcal{H}_0(\hat{f})$ denote the set of all assumptions on the joint PDF, $f$, including that any appoximations used to estimate $\tilde{w}^o$ (Eq. (10)) are eactly accurate. Let $\tilde{w} := \tilde{w}(\mathcal{H}_0(\hat{f}))$ be the corresponding estimated weights and $\alpha_1, ..., \alpha_T \in (0,1)^T$ some user-defined significance levels. If $\mathcal{H}_0(\hat{f})$ holds for the true $f$, then $P_1^{\tilde{w}}, P_2^{\tilde{w}}, ...$ are IID uniform on [0,1]:*

$$\mathbb{P}_{\mathcal{H}_0(\hat{f})}\{P_1^{\tilde{w}} \leq \alpha_1, ..., P_T^{\tilde{w}} \leq \alpha_T\} = \alpha_1 \cdots \alpha_T. \quad (12)$$

**Proof Sketch:** We defer a full proof to Appendix B. The proof builds on those for standard CTMs in Vovk (2002) and Vovk et al. (2003), which leverage the idea of "reversing time," while drawing on ideas from Tibshirani et al. (2019) and Prinster et al. (2024). We imagine that the sequence of data observations $(z_1, ..., z_T)$ is generated in two steps: first, the unordered bag of data observations, $\{z_1, ..., z_T\}$, is generated from some probability distribution; then—here generalizing Vovk (2002) and Vovk et al. (2003) by weighting permutations according to their likelihood—from all possible permutations $\sigma$ of the values $\{z_1, ..., z_T\}$, each possible sequence $(z_{\sigma(1)}, ..., z_{\sigma(T)})$ is chosen with probability proportional to $f(z_{\sigma(1)}, ..., z_{\sigma(T)})$, where $f := f_Z$ is the probability-density function[7] for the distribution $F_Z$. Roughly (ignoring borderline effects), the second step ensures that, conditionally on knowing $\{z_1, ..., z_T\}$ (and therefore unconditionally), that $P_T^{\tilde{w}^o}$ has a standard uniform distribution; when $Z_T = z_{\sigma(T)}$ is observed, this settles the value of $P_T^{\tilde{w}^o} = p_{\sigma(T)}^{\tilde{w}^o}$, and conditionally on knowing $\{z_1, ..., z_T\}$ and $Z_T = z_{\sigma(T)}$ (and therefore, after relabeling indices, on knowing $\{z_1, ..., z_{T-1}\}$), that $P_{T-1}^{\tilde{w}^o}$ also has a standard uniform distribution, and so on.

## 3.3. Main Practical Testing Objective: Testing for {Concept Shift or Unanticipated Covariate Shift}

We now provide a worked example for how weighted-conformal $p$-values can be used to test a common assumption in the robust ML literature: namely, the *covariate shift* (Shimodaira, 2000; Sugiyama et al., 2007) assumption, where the marginal input distribution $F_X$ may shift to some other distribution $G_X$ at test time, but the conditional label distribution $F_{Y|X}$ is assumed to remain invariant. In the standard covariate shift setting considered in Tibshirani et al. (2019), the oracle weights $\tilde{w}_i^o$ in Eq. (10) are proportional to density-ratio weights $g_X(X_i)/f_X(X_i)$. A common goal is to adapt to covariate shift by learning an approximate density-ratio function $\hat{w}(x) \approx g_X(x)/f_X(x)$ for reweighting the data (e.g., Sugiyama et al. (2007); Tibshirani et al. (2019); Yang et al. (2024); Zhang et al. (2024)).

However, whereas robustness results often rely on the density-ratio estimator being reasonably accurate, i.e.,

---

[6]Conservatively valid weighted CP methods would need to further set $u_{n+1} = 1$ in Eq. (9) to reduce appropriately.

[7]More generally, the Radon–Nikodym derivative.

$\widehat{w}(x) \approx g_X(x)/f_X(x)$, there is little work on *testing* for whether this reliability criterion is achieved in practice. Let $\mathbf{G}_{\mathbf{X}}^{(\widehat{\mathbf{w}})}$ denote the set of distributions for $X$ with density-ratio function $\widehat{w}(x)$ with respect to the true $F_X \in \mathbf{F}_{\mathbf{X}}$. Then, we aim to test the following joint null hypothesis:

$$\mathcal{H}_{\mathbf{0}}^{(\text{cs})} := \begin{cases} (X_i, Y_i) & \overset{iid}{\sim} \mathbf{F}_{\mathbf{Y}|\mathbf{X}} \times \mathbf{F}_{\mathbf{X}}, \ i \in [n + t_{\text{ad}} - 1] \\ (X_{n+t}, Y_{n+t}) \overset{iid}{\sim} \mathbf{F}_{\mathbf{Y}|\mathbf{X}} \times \mathbf{G}_{\mathbf{X}}^{(\widehat{\mathbf{w}})}, \ t \geq t_{\text{ad}}, \end{cases}$$
(13)

which can be roughly read as assuming that $Y \mid X$ is invariant and that $\widehat{w}(x) = g_X(x)/f_X(x)$. Thus, observing an extreme value of $p_{n+1}^{\tilde{w}}$ would convey evidence against $\mathcal{H}_0^{(\text{cs})}$, meaning that there has been a concept shift in $Y \mid X$ or that the density-ratio adaptation $\widehat{w}(x)$ is inaccurate.

### 3.4. Weighted-Conformal Test Martingales for Continual Monitoring

With Theorem 3.1 at hand, weighted-conformal test martingales (WCTMs) can be constructed from a sequence of weighted-conformal $p$-values to enable continual, anytime-valid monitoring of a customizeable null hypothesis $\mathcal{H}_0(\hat{f})$. Just as in the construction of standard CTMs (Section 2.3), WCTMs require a betting function $h$ (we assume the betting function in Eq. (6)) and a strategy $\nu$ for placing bets $\epsilon_t$ at each time $t$ that can only depend on past $p$-value observations (we use the composite jumper strategy from Vovk et al. (2022)). Then, a WCTM is constructed by feeding a sequence of weighted-conformal $p$-values $p_1^{\tilde{w}}, ..., p_t^{\tilde{w}}, ...$ into Eq. (7) (i.e., by setting $p_i := p_i^{\tilde{w}}$ in Eq. (7)). Due to the $P_t^{\tilde{w}}$ being IID uniform on [0,1] under $\mathcal{H}_0(\hat{f})$ (Theorem 3.1), the stochastic process $\tilde{M}_0, \tilde{M}_1, ..., \tilde{M}_t$ is a nonnegative test martingale for $\mathcal{H}_0(\hat{f})$, and by Ville's inequality (Ville, 1939) it achieves the anytime-valid control over false alarms in the following proposition (proof in Appendix B).

**Proposition 3.2.** *(WCTM anytime-valid false-alarm control)* *Let* $\tilde{M}_0, \tilde{M}_1, ..., \tilde{M}_t, ...$ *be a WCTM constructed from the sequence of weighted-conformal $p$-values* $p_1^{\tilde{w}}, p_2^{\tilde{w}}, ..., p_t^{\tilde{w}}, ....$ *Then, assuming* $\mathcal{H}_0(\hat{f})$,

$$\mathbb{P}_{\mathcal{H}_0(\hat{f})}\big(\exists \, t : \tilde{M}_t/\tilde{M}_0 \geq c\big) \leq 1/c.$$
(14)

### 3.5. Scheduled or Multistage WCTM-Based Monitoring

The anytime-valid guarantee in Eq. (14) controls the probability of raise a false alarm at *any* time in an infinite sequence of observations, but this can be overly strong, especially when one has a set time horizon in mind for monitoring. For such "scheduled" or "multistage" monitoring, we improve the efficiency (speed in detecting true shifts) by following Vovk et al. (2021); Vovk (2021) and augmenting WCTMs with standard changepoint detection metrics such as CUSUM (Page, 1954) and Shiryaev-Roberts (Roberts,

1966; Shiryaev, 1963) while achieving a weaker form of validity. Consider the Shiryaev-Roberts procedure applied to WCTMs $\tilde{M}_t$, whose $k$-th stage alarm time is

$$\tau_k := \min \Big\{ t > \tau_{k-1} : \sum_{i=\tau_{k-1}}^{t-1} \frac{\tilde{M}_t}{\tilde{M}_i} \geq c \Big\}, \ k \in \mathbb{N}. \quad (15)$$

This Shiryaev-Roberts procedure based on standard CTMs controls the average run length (ARL) under $\mathcal{H}_0(\hat{f})$, where the procedure is expected to be reset after $c$ timesteps.

**Proposition 3.3.** *(WCTM-based Shiryaev-Roberts ARL control)* *Let* $\tau_1$ *denote the Shiryaev-Roberts stopping time (Eq. (15)) based on a WCTM* $\tilde{M}_0, \tilde{M}_1, ..., \tilde{M}_t, ...$ *constructed for the null hypothesis* $\mathcal{H}_0(\hat{f})$. *Then, assuming* $\mathcal{H}_0(\hat{f})$,

$$\mathbb{E}_{\mathcal{H}_0(\hat{f})}[\tau_1] \geq c.$$
(16)

### 3.6. WCTM Implementation in WATCH: Dynamic Online Adaptation and Root-Cause Analysis

All our implementations and experiments in this paper focus on WCTMs that continuously adapt to mild covariate shifts (e.g., Fig 1a) while raising alarms in response to either extreme covariate shifts (e.g., Fig 1b) or concept shifts in $Y|X$ (e.g., Fig 1c). These goals correspond to monitoring the $\mathcal{H}_0^{(cs)}$ null hypothesis given in Eq. (13). Our adaptation procedure performs online density ratio estimation with an online probabilistic classifier (e.g., a multilayer perceptron or logistic regression model similar to in Zhang et al. (2024)) to update the estimation of $\widehat{w}^{(t)}(x)$ at each timestep $t$.

**Online Adaptation with Dynamic Initialization** For the initial adaptation stage, a key question is *when* to begin the adaptation procedure, or when to begin estimating $\widehat{w}^{(t)}(x)$. Our methods make this determination automatically and dynamically by running a secondary standard CTM that is restricted to only monitoring for changepoints in the marginal $X$ distribution via a nearest-neighbor nonconformity score (Vovk et al., 2021). At deployment, the main WCTM method is initially a standard CTM (with uniform weights); then, once the secondary "$X$-CTM" method exceeds a pre-determined adaptation threshold (see gray paths in Fig 1) at some time $t_{\text{ad}}$, this triggers the adaptation of the main WCTM monitoring method (blue paths in Fig 1) and corresponding weighted CP intervals. For efficiency purposes, after the adaptation time $t_{\text{ad}}$, the CP calibration set $[n + t_{\text{ad}}]$ is treated as fixed (no longer adding test points to it online), meaning the weighted-conformal $p$-values are computed summing only over $[n + t_{\text{ad}} - 1] \cup \{n + t\}$,

$$p_{n+t}^{\tilde{w}} = \sum_{i \in [n+t_{\text{ad}}-1] \cup \{n+t\}} \tilde{w}_i^{(t)} \begin{bmatrix} \mathbb{1}\{v_i > v_{n+t}\} \\ +u_{n+t}\mathbb{1}\{v_i = v_{n+t}\} \end{bmatrix},$$
(17)

where here $\tilde{w}_i^{(t)} \propto \widehat{w}^{(t)}(X_i)$. See Appendix C for details.

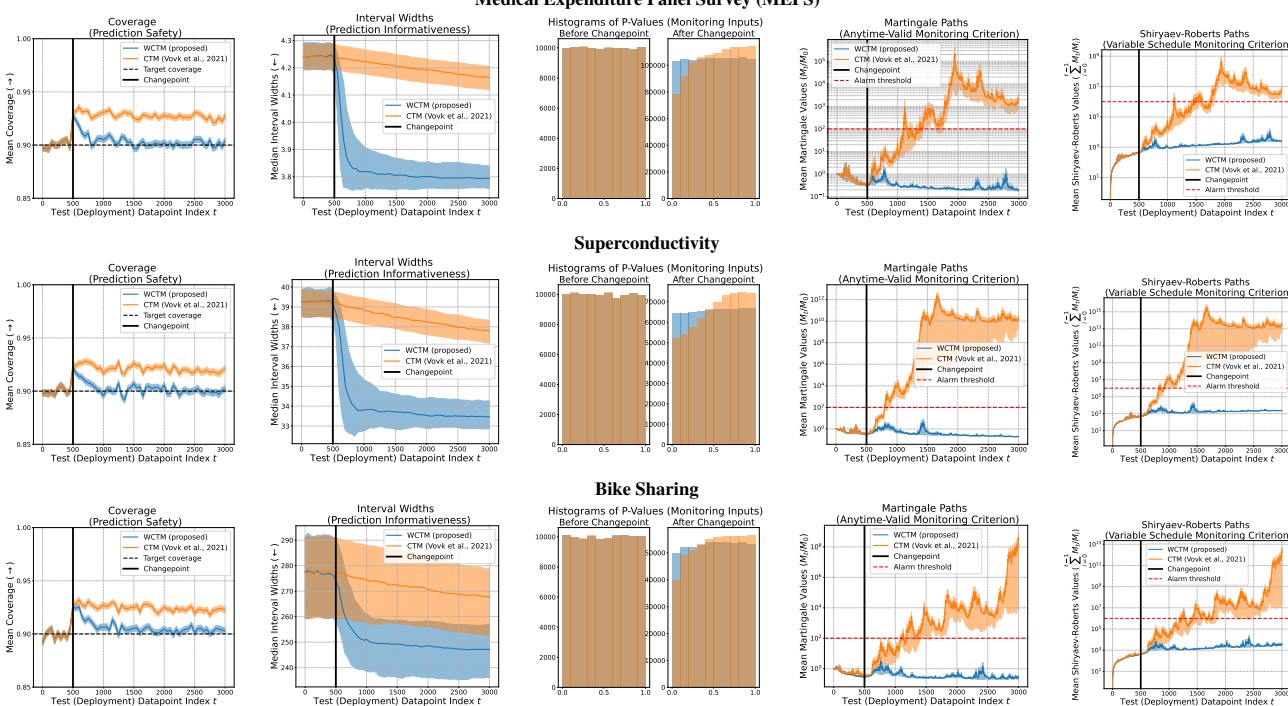

Figure 2: Tabular data results for the benign covariate shift setting to evaluate the adaptation ability of proposed WCTM methods (blue); all values are averaged over 200 random seeds. Training and calibration sets were sampled uniformly at random (with 1/3 of the total data used for training and calibration each), while post-changepoint test-set datapoints were bias-sampled from the remaining holdout data with probability proportional to $\exp(\lambda \cdot h(x))$. The shift-magnitude scalar $\lambda$ for each dataset was set as $\lambda_{\text{mep}} = 5.0$, $\lambda_{\text{sup}} = 2.5$, and $\lambda_{\text{bik}} = 5.0$. The function $h$ was selected to simulate realistic shifts as described in the main text. Error regions represent standard errors for coverage, martingale paths, ans Shiryaev-Roberts paths, and interquartile range for interval widths. Whereas standard CTMs (orange) raise unnecessary alarms with their anytime-valid and scheduled monitoring criteria, WCTMs avoid doing so by adapting online to the shift. That is, WCTMs maintain target coverage, adapt by increasing interval sharpness, and avoid unneeded alarms.

**Root-Cause Analysis with Parallelized WCTM and $X$-CTM** The parallel implementation of both the primary WCTM and the secondary $X$-CTM method furthermore enables root-cause analysis, that is to diagnose whether performance degradation was due to a harmful covariate shift (e.g., Fig 1b) or a fundamental concept shift (e.g., Fig 1c). That is, if both the WCTM and the $X$-CTM have detected changepoints, then a harmful shift can be diagnosed as an extreme covariate shift (e.g., Fig 1b); on the other hand, if the $X$-CTM does not detect a changepoint, this suggests that no covariate shift is present, and the WCTM alarm was instead due to a concept shift in $Y|X$ (Figure 1).

## 4. Experiments

We conduct a comprehensive empirical analysis of the WATCH framework on real-world datasets with various distribution shifts. Our results show that WATCH adapts effectively to benign shifts (Section 4.1) and triggers alarms when it fails to adapt (Section 4.2), while also quickly detecting harmful shifts with little delay (Section 4.3). Details on the datasets, models, and additional results, can

be found in Appendix E. Code to reproduce all experiments is available at the following repository: https://github.com/aaronhan223/watch.

**Baselines** We compare the proposed WCTM methods that constitute WATCH against standard CTMs (Vovk et al., 2021) in all experiments. To evaluate detection speed on true harmful shifts (sec 4.3), we also compare against methods for directly performing sequential hypothesis testing and changepoint detection on the set-prediction miscoverage risk, as proposed by Podkopaev and Ramdas (2021b).

In all experiments and for all baselines, the underlying ML predictor being monitored was a neural network. On the tabular data, we used the scikit-learn (Pedregosa et al., 2011) MLPRegressor (with L-BFGS solver and logistic activation); for the image data, we used a 3-layer MLP with ReLU activations on the MNIST datasets and a ResNet-32 (He et al., 2016) on CIFAR-10 datasets. For weight estimation, we use a 3-layer MLP with ReLU activations to distinguish between source and target distributions.

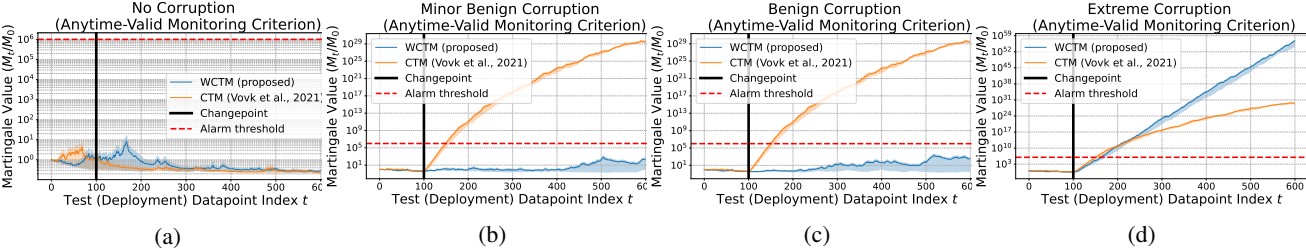

Figure 3: Example martingale trajectories of WCTM and CTM on CIFAR-10-C with increasing levels of corruption. WCTM adapt to milder shifts to avoid unnecessary alarms while still detecting severe shifts. All plots are averaged from 5 random experiments.

Table 1: Average detection delay (ADD) computed over 200 random seeds for each monitoring method and dataset (error bars are standard errors). Anytime-valid monitoring methods are WCTMs (proposed), CTMs (Vovk, 2021), and sequential testing from Podkopaev and Ramdas (2021b) (PR-ST); multistage monitoring methods are Shiryaev-Roberts (SR) procedure applied to WCTMs (proposed), SR applied to CTMs (Vovk, 2021), changepoint detection on the miscoverage risk from Podkopaev and Ramdas (2021b) in both online (PR-CD-online) and minibatched (PR-CD-50) variants. Results corresponding to our method are in blue. The best ADD are **bolded**.

| | *Monitoring Criterion* ($\rightarrow$) | *Anytime-Valid Criterion* | | *Scheduled, Multistage Criterion* | | |
|---|---|---|---|---|---|---|
| | *Monitoring Method* ($\rightarrow$) | (W)CTM | PR-ST | SR via (W)CTM | PR-CD-online | PR-CD-50 |
| *Dataset* ($\downarrow$) | *CP Type* ($\downarrow$) | **ADD ($\pm$ SE)** | **ADD ($\pm$ SE)** | **ADD ($\pm$ SE)** | **ADD ($\pm$ SE)** | **ADD ($\pm$ SE)** |
| **MEPS** | Weighted | 115.8 ($\pm$3.8) | 536.6 ($\pm$23.9) | 102.1 ($\pm$3.2) | 111.3 ($\pm$3.8) | 196.5 ($\pm$5.4) |
| | Standard | 114.8 ($\pm$3.7) | 475.2 ($\pm$18.8) | 101.8 ($\pm$3.2) | 111.0 ($\pm$3.9) | 194.5 ($\pm$5.3) |
| **Bike Sharing** | Weighted | 190.0 ($\pm$8.6) | 602.5 ($\pm$24.7) | 155.6 ($\pm$6.6) | 154.3 ($\pm$5.9) | 241.2 ($\pm$6.5) |
| | Standard | 192.4 ($\pm$8.9) | 607.7 ($\pm$24.7) | 157.4 ($\pm$6.6) | 157.4 ($\pm$5.9) | 244.3 ($\pm$6.6) |
| **Superconduct** | Weighted | 173.2 ($\pm$6.2) | 802.9 ($\pm$30.5) | 142.8 ($\pm$5.0) | 192.9 ($\pm$6.0) | 289.1 ($\pm$6.1) |
| | Standard | 169.7 ($\pm$5.8) | 813.1 ($\pm$29.9) | 142.3 ($\pm$5.0) | 187.6 ($\pm$5.8) | 287.9 ($\pm$6.1) |

## 4.1. WCTMs Adapt to Mild and Benign Shifts to Avoid Unnecessary Alarms

**WCTMs Adapt to Benign Covariate Shifts** Figure 2 compares the average performance (across 200 random seeds) of WCTMs to standard CTMs across 3000 sequentially-observed test datapoints, where the true changepoint shift is induced after the 500th test point. From the conformal coverage plots in the first column, it is clear that all the shifts are benign in that coverage for the corresponding (standard or weighted) CP methods—a metric of prediction safety—does not degrade below the target level (0.9) after the changepoint. In fact, for the baseline method (orange), coverage *increases*, which could be considered a "beneficial" shift; nonetheless, the standard CTM raises unnecessary alarms across all datasets, for both its anytime-valid (fourth column) and scheduled (fifth column) monitoring criteria. In contrast, the proposed WCTM (blue) avoids these unnecessary alarms across all datasets and both monitoring criteria. That is, the WCTM *adapts* to the benign covariate shift by decreasing its interval widths (second column)—indicating more informative predictions—while maintaining target coverage. The relatively uniform distribution of the postchangepoint weighted $p$-values (third column, blue) is empirical evidence validating Theorem 3.1; meanwhile, the martingale (fourth column) and Shiryaev-Roberts (fifth column) WCTM paths avoiding alarms supports Theorems 3.2 and 3.3 respectively.

## 4.2. From Mild to Extreme Covariate Shifts: WCTMs Raise Alarms if Unable to Adapt

We demonstrate such a property of WCTM using image corruption experiments. Shift severity was controlled by increasing the proportion of corrupted samples in the test set relative to training and calibration. Figure 3 illustrates the behavior of WCTM in comparison to CTM on CIFAR-10 under varying levels of brightness corruption. When the target distribution is completely clean (a), neither method triggers a false alarm. We then introduce level-1 brightness corruption while retaining over $50\%$ clean samples (b), creating a very mild, benign shift. In this scenario, CTM quickly raises an unnecessary alarm, whereas WCTM successfully avoids it. Next, we increase the corruption to level-3 and reduce the clean-sample ratio to $40\%$ (c). Although WCTM reacts to this shift, it still does not trigger an alarm. Finally, at corruption level-5, with the clean-sample ratio reduced to $30\%$ (d), WCTM does raise the alarm. See Appendix D.6 for further experiments on corrupted image data and D.2 for further analysis and explanation of what characterizes mild, moderate, or extreme covariate shifts.

## 4.3. WCTMs Detect Harmful Concept Shifts Faster than Sequentially Tracking Loss Metrics

Because concept shifts fundamentally change the $Y|X$ relationship, they are generally harmful, and monitoring meth-

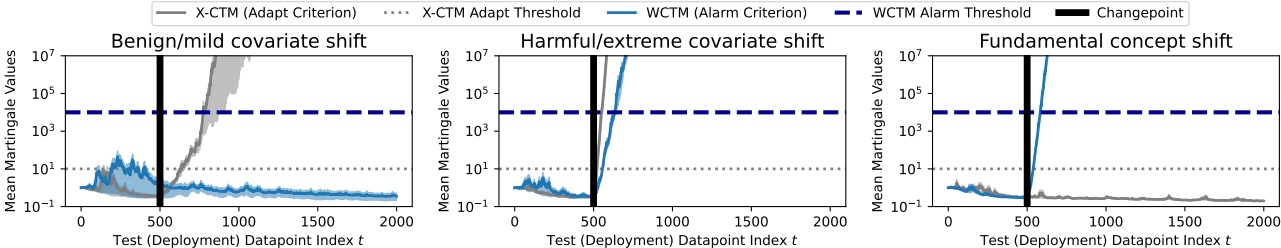

Figure 4: Results for root-cause analysis with a WCTM (blue) and a secondary $X$-CTM (gray) on the MEPS dataset, averaged over 200 random seeds. The WCTM and $X$-CTM together are able to diagnose different types of shifts.

ods should detect them as quickly as possible. Our experiments compare the average detection delay (ADD) of WCTM methods relative to comparable CTM methods, as well as against methods for sequentially tracking the loss metrics directly with comparable false-alarm control. That is, the sequential testing procedures proposed in Podkopaev and Ramdas (2021b) (PR-ST) have anytime-valid control comparable to (W)CTMs, while the changepoint detection procedure in Podkopaev and Ramdas (2021b) (PR-CD) has average-run-length control comparable to running the Shiryaev-Roberts procedure (15) on top of (W)CTMs. Both PR-ST and PR-CD methods can be used to monitor the miscoverage risks of either weighted CP or standard CP methods, though they are less data efficient (requiring an extra holdout set for computing concentration inequalities). We compared to the betting-based -process in Podkopaev and Ramdas (2021b) (with $\epsilon_{\text{tol}} = 0$) because those methods were reported to perform the best and are more comparable to (W)CTMs, which are also betting-based. We compared WCTMs to sequential testing variant with standardized anytime-false alarm rate of 0.01; for SR-WCTMs we compared vs changepoint detection variant with common average-run-length (under the null) of 20,000.

Table 1 reports the average detection delay (ADD) results for monitoring methods grouped by false-alarm control type, with proposed methods in blue. Among the anytime-valid monitoring methods, WCTMs and CTMs achieve comparably fast ADD; however, relative to comparable PR-ST methods, (W)CTMs are over *three times* faster. Meanwhile, among the stagewise monitoring methods, WCTM and CTM-based SR procedures are comparable, but with significantly faster ADD than PR-CD run either online or in minibatches. PR-CD-online has lower ADD than in minibatches, but its wall-clock runtime is far slower (Appendix D.3) due to the method's $\mathcal{O}(t^2)$ time complexity, whereas the proposed WCTM and SR-WCTM methods are $\mathcal{O}(t)$.

### 4.4. WCTMs Diagnose Root Cause of Harmful Shifts

The left plot corresponds to a benign covariate shift setting (same setting as in Figure 2), and WATCH diagnoses it as

such with the $X$-CTM achieving a large value while the WCTM adapts to the shift, avoiding an alarm. The middle plot corresponds to an extreme covariate shift setting, and WATCH diagnoses it by the $X$-CTM identifying covariate shift and the WCTM raising an alarm due to potential harm. Lastly, the right plot corresponds to a harmful concept shift, and WATCH identifies it as such by the WCTM raising an alarm, but with the $X$-CTM maintaining lower values, suggesting that no covariate shift has occurred.

## 5. Summary and Future Directions

In this paper we introduced the novel methods of weighted-conformal test martingales (WCTMs), which are constructed from sequences of weighted-conformal $p$-values. We presented new theory for how online weighted-conformal $p$-values can be used in sequential hypothesis testing, and we demonstrated how WCTMs enable continual monitoring of deplyed AI/ML models. The proposed approach, WATCH, achieves three main goals: (1) *adaptation* to benign shifts to avoid unnecessary alarms and improve the utility of CP prediction sets for end-users; (2) *fast detection* of harmful shifts; and (3) *root-cause analysis* to identify the cause of any performance degradation as a harmful covariate shift or a fundamental concept shift.

Our contribution opens up several promising directions for future work. Future theory directions include developing specific WCTM algorithms for testing other nonparametric null hypotheses, exploring connections to conditional permutation tests (e.g., Berrett et al. (2020)), analyzing WCTMs' efficiency with respect to certain alternative hypotheses, developing robustness results for approximate weights, and finer-grained root-cause analysis. For applications, implementations of WATCH to monitor the risks of AI systems in real practical settings such as healthcare and extensions to monitor foundation models and AI agents would be valuable. We hope that WATCH encourages greater attention on post-deployment monitoring for safer AI systems, with a focus on adaptive monitoring, fast detection of harmful shifts, and diagnostics to inform recovery.

## Impact Statement

This paper presents research aimed at propelling advancements in the broad domain of machine learning. The implications of our findings are wide-ranging, with potential high-stake applications in sectors including healthcare, autonomous driving, and e-commerce. Based on our current understanding, this research does not warrant an ethics review, and a detailed discussion of the potential societal impacts is not required at the current stage.

## Acknowledgements

D.P. was funded by a PhD fellowship from the Amazon Initiative for Interactive AI (AI2AI); D.P., X.H., A.L., and S.S. were partially funded by the Gordon and Betty Moore Foundation grant #12128; D.P., X.H., and S.S. were partially funded by National Science Foundation (NSF) grant #1840088. A.L. is also supported by an Amazon Research Award. The authors would like to thank Yaniv Romano for helpful discussions on a draft of this paper; Jwala Dhamala and Jie Ding for helpful discussions throughout the JHU + AI2AI collaboration; and the anonymous ICML reviewers for informative feedback.

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

# Appendix for
# "WATCH: Adaptive Monitoring for AI Deployments via Weighted-Conformal Martingales"

## A. Related Works

This work is motivated by developing methods for monitoring AI/ML deployments that perform three key functions: (1) online adaptation to mild or benign data shifts; (2) rapid detection of extreme or harmful shifts that necessitate updates; and (3) identifying the root-cause of degradation to inform appropriate recovery. Although these monitoring goals could be viewed from many perspectives, in discussing related work we primarily focus on methods in anytime-valid inference, sequential testing of nonparametric null hypotheses, and especially conformal test martingales, which most closely related to our own.

**Sequential testing for changepoints in the data distribution:** Sequential hypothesis testing to detect changes in the data distribution is an old and widely-studied problem, dating at least to Wald's sequential probability ratio test (Wald, 1945). However, classic sequential changepoint detection methods often required a prespecified stopping time and were primarily designed for testing simple, often parametrically-specified null hypotheses—for an overview of classic parametric methods, we defer to a relevant textbook Tartakovsky et al. (2014) and review articles (Veeravalli and Banerjee, 2014; Xie et al., 2021). In contrast, our weighted-conformal test martingale (WCTM) methods belong to a more recent literature called sequential anytime-valid inference (SAVI), which allow for arbitrary stopping times, and specifically our work is situated in the recent SAVI literature on testing composite and nonparametric null hypotheses. We refer readers to Ramdas et al. (2023) for a recent review of the SAVI literature, as well as to the textbook by Shafer and Vovk (2019) on the closely related topic of testing-by-betting.

**Standard conformal test martingales:** Within the SAVI literature, our work is most closely related to conformal test martingales (CTMs), which are martingales constructed from a sequence of standard conformal $p$-values for continually testing the assumption that the data are exchangeability or independent and identically distributed (IID). Our WCTM methods generalize standard CTMs for testing a broader range of null hypotheses, including those where one wishes to accommodate or adapt to certain anticipated changes in the data distribution (e.g., adapting to mild covariate shifts). Standard CTMs were initially introduced by Vovk et al. (2003), while drawing on theory for the calibration of online conformal prediction methods developed in Vovk (2002). Since then, various works have further developed standard CTMs, such as by introducing new betting and score functions for more efficient changepoint detection, ensembling CTMs, and demonstrating their performance on real-world applications (Ho, 2005; Fedorova et al., 2012; Volkhonskiy et al., 2017; Ho et al., 2019; Vovk et al., 2021; Vovk, 2021; Eliades and Papadopoulos, 2022; 2023). CTMs are also discussed in textbooks on conformal prediction (Vovk et al., 2005; Angelopoulos et al., 2024). The online form of conformal prediction that CTMs are based on is one form of online compression model, as defined and discussed in Vovk (2003); Vovk et al. (2005); Vovk (2023).

In particular, Volkhonskiy et al. (2017) developed inductive CTMs (i.e., CTMs based on inductive or split conformal (Papadopoulos et al., 2002; Papadopoulos, 2008)) with score and betting functions specifically taylored to fast changepoint detection; Vovk et al. (2021) proposed an approach to using standard CTMs to monitor for when a deployed AI/ML system should be retrained; and Vovk (2021) provided a comprehensive and detailed review of CTMs as methods for testing the IID or exchangeability assumptions. Bar et al. (2024) implement methods based on CTMs for test time-adaptation of a classifier's point prediction and Hindy et al. (2024) leverage multiple CTMs over different feature spaces to aid in diagnostic runtime monitoring. Several works have developed or used CTMs for testing for concept shift (i.e., shift in $Y \mid X$) (Ho, 2005; Eliades and Papadopoulos, 2022; 2023; Vovk, 2020), but all of these have focused on a classification setting with a limited number of classes (e.g., to implement label-conditional CTMs, or using standard CTMs that lack ability to disambiguate between covariate and concept shifts). In contrast, our specific WCTMs implemented in this paper are able to test for concept shift in both regression and classification settings while also being able to disambiguate between concept shifts and extreme covariate shifts with the aid of an additional $X$-CTM.

**Confidence sequences, $e$-processes, and other multiple-hypothesis testing methods:** Our work also relates more broadly to other SAVI and testing-by-betting methods outside of CTMs such as confidence sequences and $e$-processes—we refer to the review paper of Ramdas et al. (2023) for full exposition of these topics. In particular, in the main paper we empirically compared our WCTM methods against the betting-based sequential testing and changepoint detection methods in Podkopaev and Ramdas (2021b) (the theory for which was developed in Waudby-Smith and Ramdas (2024)) because this paper is the

closest to ours in its motivation of monitoring deployed AI/ML deployments among (non-CTM) SAVI papers. As we note in the main paper, for monitoring the risk of a set-valued (e.g., conformalized) AI/ML predictor, our WCTM methods are generally more data-efficient (not requiring separate datasets for conformalization and testing, as Podkopaev and Ramdas (2021b) does); our WCTM-based Shiryaev-Roberts procedure is more computationally efficient than the compareable changepoint detection method in Podkopaev and Ramdas (2021b) (i.e., $\mathcal{O}(t)$ versus $\mathcal{O}(t^2)$); and, empirically we found that our methods often detect harmful concept shifts faster than comparable methods in Podkopaev and Ramdas (2021b). More recently, I Amoukou et al. (2024) built on Podkopaev and Ramdas (2021b) by developing similar methods for when ground-truth labels do not become available and need to be estimated.

A key target of the SAVI literature on testing composite or nonparametric null hypotheses has been developing new approaches for testing the IID or exchangeability assumptions. Other than CTMs, the other main nontrivial approach to sequentially testing exchangeability was developed in Ramdas et al. (2022) based on the ideas of universal inference (Wasserman et al., 2020); more recently, other approaches have emerged based on pairwise betting (Saha and Ramdas, 2024) and sequential Monte Carlo testing (Fischer and Ramdas, 2025), the latter of which can be viewed as a special case of CTMs (Vovk, 2021) streamlined for a particular alternative hypothesis. Otherwise, there are various and proliferating other methods for sequential nonparametric changepoint detection (e.g., Shin et al. (2022); Shekhar and Ramdas (2023a;b;c); Podkopaev and Ramdas (2023)). SAVI and testing-by-betting methods are also being leveraged for a wide variety of applications including interpretability (Teneggi and Sulam, 2024), conditional independence testing (Shaer et al., 2023), applications in finance (Shafer and Vovk, 2019), and more.

Testing-by-betting, which is fundamental to SAVI (Ramdas et al., 2023), can be understood as one approach to multiple hypothesis testing that is especially advantageous in sequential settings. Other related works that leverage conformal prediction for hypothesis testing while accounting for multiple-testing corrections, but in batch settings, include Bates et al. (2023); Bashari et al. (2023); Vovk and Wang (2023); Gauthier et al. (2025); Lee and Ren (2025).

**Weighted conformal prediction and adapting to distribution shifts:** Weighted conformal prediction (e.g., Tibshirani et al. (2019); Podkopaev and Ramdas (2021a); Xu and Xie (2021); Fannjiang et al. (2022); Prinster et al. (2022; 2023); Stanton et al. (2023); Barber et al. (2023); Farinhas et al. (2023); Nair and Janson (2023); Yang et al. (2024); Feldman and Romano (2024); Barber and Tibshirani (2025)) is broadly an approach to proactively adapting the validity of conformal predictive sets to distribution shift by reweighting nonconformity scores using either knowledge or estimates of the distribution shift. Any weighted CP prediction set is associated with a weighted-conformal $p$-values, as described in the main paper; accordingly, a WCTM can be constructed on top of any weighted CP method deployed on a sequence of data observations to continually monitor the assumptions or approximations underlying that WCP method's implementation. There are of course many other approaches to adapting to distribution shifts at test time; for example, one work that is similar to ours with regard to this motivation, but that does not fall under weighted CP, is Bar et al. (2024), which leverages standard CTMs to guide the test-time adaptation of a classifier's point prediction by entropy matching.

# B. Proof of Theorems

## B.1. Proof for Theorem 3.1 (Weighted Conformal $p$-value Validity)

The proof can be viewed as a generalization of the proofs for Theorem 1 in Vovk et al. (2003) and Theorem 2 in Vovk (2002), while drawing on analysis from Tibshirani et al. (2019) and exposition and discussion from Prinster et al. (2024). The key difference relative to Vovk (2002) and Vovk et al. (2003) is that, whereas those papers use the assumption of exchangeability to place equal weight on every permutation of the data observations, here we avoid this assumption by first proving a general result for an arbitrary (potentially non-exchangeable) joint distribution. We then describe how this implies that the validity of more specific and tractable methods is premised on the assumptions or approximations used for practical implementation.

The basic idea for the proof begins with setup from Vovk (2002), for "reversing time." In particular, we imagine that the sequence of data observations $(z_1, ..., z_T)$ is generated in two steps: first, the unordered bag or multiset of data observations, $\{z_1, ..., z_T\}$, is generated from some probability distribution (that is, the image of $F_Z$ under the mapping $(z_1, z_2, ...) \rightarrow \{z_1, ..., z_T\}$); then—here generalizing Vovk (2002) and Vovk et al. (2003) by weighting permutations according to their likelihood—from all possible permutations $\sigma$ of the values $\{z_1, ..., z_T\}$, each possible sequence $(z_{\sigma(1)}, ..., z_{\sigma(T)})$ is chosen with probability proportional to $f(z_{\sigma(1)}, ..., z_{\sigma(T)})$, where $f := f_Z$ is the probability-density function[8] for the distribution $F_Z$. Roughly (ignoring borderline effects), the second step ensures that, conditionally on knowing $\{z_1, ..., z_T\}$ (and therefore

---

[8] More generally, the Radon–Nikodym derivative.

unconditionally), that $P_T^{\tilde{w}^o}$ has a standard uniform distribution; when $Z_T = z_{\sigma(T)}$ is observed, this settles the value of $P_T^{\tilde{w}^o} = p_{\sigma(T)}^{\tilde{w}^o}$, and conditionally on knowing $\{z_1, ..., z_T\}$ and $Z_T = z_{\sigma(T)}$ (and therefore, after relabeling indices, on knowing $\{z_1, ..., z_{T-1}\}$), that $P_{T-1}^{\tilde{w}^o}$ also has a standard uniform distribution, and so on.

**Lemma B.1.** *For any trial $t$ and any confidence level $\alpha \in (0,1)$,*

$$\mathbb{P}\{P_t^{\tilde{w}^o} \leq \alpha \mid E_z^{(t)}\} = \alpha. \tag{18}$$

*Proof of Lemma B.1.* We begin by conditioning on the event $\{Z_1, ..., Z_t\} = \{z_1, ..., z_t\}$, which we denote as $E_z^{(t)}$, and we consider drawing any particular ordering or permutation $\sigma$ of the data values with probability according to $f$, that is with probability proportional to $f(z_{\sigma(1)}, ..., z_{\sigma(t)})$.

Note that for any $i \in [t]$, if a permutation is drawn such that $\sigma(t) = i$, this means that $Z_t = z_{\sigma(t)} = z_i$; moreover, because the score function $\hat{\mathcal{S}}$ is bijective, this further implies that $V_t = v_{\sigma(t)} = v_i$. Thus, given the bag of data $E_z^{(t)}$, recall that for each $i \in [t]$, the probability of drawing such a permutation is given by the "oracle weights"

$$\tilde{w}_i^o := \mathbb{P}\{V_t = v_i \mid E_z^{(t)}\} = \mathbb{P}\{Z_t = z_i \mid E_z^{(t)}\} = \frac{\sum_{\sigma:\sigma(t)=i} f(z_{\sigma(1)}, ..., z_{\sigma(t)})}{\sum_\sigma f(z_{\sigma(1)}, ..., z_{\sigma(t)})}, \tag{19}$$

which we assume to be well-defined, which will generally be the case in practice, where at least $f(z_1, ..., z_T) > 0$ for the true (identity permutation) ordering of the data observations $(z_1, ..., z_T)$.

This implies that the distribution of $V_t \mid E_z^{(t)}$, the conditional distribution of the test-point score given the bag of data values $E_z^{(t)}$, is given by

$$V_t \mid E_z^{(t)} \sim \sum_{i=1}^t \tilde{w}_i^o \cdot \delta_{v_i}.$$

For any $i \in [t]$, define a conservative WCP $p$-value, $p_i^{\tilde{w}^o+}$, and an anticonservative WCP $p$-value, $p_i^{\tilde{w}^o-}$, as

$$p_i^{\tilde{w}^o+} := \sum_{j=1}^t \tilde{w}_j^o \cdot \mathbb{1}\{v_j \geq v_i\}$$

$$p_i^{\tilde{w}^o-} := \sum_{j=1}^t \tilde{w}_j^o \cdot \mathbb{1}\{v_j > v_i\}.$$

It is worth noting that $p_t^{\tilde{w}^o+}$ is a valid $p$-value for $f$: $\mathbb{P}\{P_t^{\tilde{w}^o+} \leq \alpha \mid E_z^{(t)}\} \leq \alpha \implies \mathbb{P}\{P_t^{\tilde{w}^o+} \leq \alpha\} \leq \alpha$. However, the lemma claims *exact* validity for $p_t^{\tilde{w}^o}$ conditional on $E_z^{(t)}$, which we will now proceed to show.

Observe that for all $i \in [t]$, $p_i^{\tilde{w}^o-} < p_i^{\tilde{w}^o+}$ and

$$p_i^{\tilde{w}^o+} - p_i^{\tilde{w}^o-} = \sum_{j=1}^t \tilde{w}_j^o \cdot \mathbb{1}\{v_j = v_i\}.$$

Moreover, observe that as in the proof for Lemma 1 in Vovk (2002), the semi-closed intervals $[p_i^{\tilde{w}^o-}, p_i^{\tilde{w}^o+})$ either coincide or are disjoint, and $\cup_{i=1}^t [p_i^{\tilde{w}^o-}, p_i^{\tilde{w}^o+}) = [0, 1)$.

Similarly as in the proof for Lemma 1 in Vovk (2002), for an $\alpha \in (0, 1)$, let us partition the indices as follows, where we say that an index $i$ is

- "strange" if $p_i^{\tilde{w}^o+} \leq \alpha$,

- "ordinary" if $p_i^{\tilde{w}^o-} > \alpha$,

- and "borderline" if $p_i^{\tilde{w}^o-} \leq \alpha < p_i^{\tilde{w}^o+}$.

Let $i'$ denote the index of any borderline example, and denote $p^{\tilde{w}^o+} := p_{i'}^{\tilde{w}^o+}$ and $p^{\tilde{w}^o-} := p_{i'}^{\tilde{w}^o-}$. Then, the probability (conditional on $E_z^{(t)}$, drawing each permutation $\sigma$ with probabilities according to $f$) that the last index $\sigma(t)$ is strange is $p^{\tilde{w}^o-}$; the probability that $\sigma(t)$ is ordinary is $1 - p^{\tilde{w}^o+}$; and, the probability that $\sigma(t)$ is borderline is $p^{\tilde{w}^o+} - p^{\tilde{w}^o-}$. Moreover, observe that if $\sigma(t)$ is strange, then $p_{\sigma(t)}^{\tilde{w}^o} \le \alpha$ (by definition) and if $\sigma(t)$ is borderline then the event that $p_{\sigma(t)}^{\tilde{w}^o} \le \alpha$ is determined by the independent uniform $u_t$, and thus the probability of this event is $\frac{\alpha - p^{\tilde{w}^o-}}{p^{\tilde{w}^o+} - p^{\tilde{w}^o-}}$. That is,

$$
\begin{aligned}
\mathbb{P}\{P_t^{\tilde{w}^o} \le \alpha \mid E_z^{(t)}\} &= \mathbb{P}\{P_t^{\tilde{w}^o} \le \alpha \mid E_z^{(t)}, \ \sigma(t) \text{ is strange}\} \cdot \mathbb{P}\{\sigma(t) \text{ is strange} \mid E_z^{(t)}\} \\
&\quad + \mathbb{P}\{P_t^{\tilde{w}^o} \le \alpha \mid E_z^{(t)}, \ \sigma(t) \text{ is ordinary}\} \cdot \mathbb{P}\{\sigma(t) \text{ is ordinary} \mid E_z^{(t)}\} \\
&\quad + \mathbb{P}\{P_t^{\tilde{w}^o} \le \alpha \mid E_z^{(t)}, \ \sigma(t) \text{ is borderline}\} \cdot \mathbb{P}\{\sigma(t) \text{ is borderline} \mid E_z^{(t)}\} \\
&= p^{\tilde{w}^o-} + 0 + (p^{\tilde{w}^o+} - p^{\tilde{w}^o-}) \cdot \frac{\alpha - p^{\tilde{w}^o-}}{p^{\tilde{w}^o+} - p^{\tilde{w}^o-}} \\
&= \alpha.
\end{aligned}
$$

$\square$

With Lemma 1 in hand, we can now proceed with the proof for the main theorem. Temporarily fix a positive integer $T$; following the strategy of Vovk et al. (2003), we will first prove by induction that for any $t = 1, ..., T$ and any $\alpha_1, ..., \alpha_t \in [0,1]^t$, that

$$
\mathbb{P}\{P_t^{\tilde{w}^o} \le \alpha_t, ..., P_1^{\tilde{w}^o} \le \alpha_1 \mid E_z^{(t)}\} = \alpha_t \cdots \alpha_1. \tag{20}
$$

For $t = 1$, Eq. (20) immediately follows from Lemma 1. For $t > 1$, by the law of total probability over $\sigma(t)$ (i.e., over the last index value after drawing a permutation $\sigma$ from $E_z^{(t)}$), the fundamental bridge between probability and expectation, and properties of the indicator function, we have

$$
\begin{aligned}
&\mathbb{P}\{P_t^{\tilde{w}^o} \le \alpha_t, ..., P_1^{\tilde{w}^o} \le \alpha_1 \mid E_z^{(t)}\} \\
&= \sum_{\sigma(t)=1}^{t} \mathbb{P}\{P_t^{\tilde{w}^o} \le \alpha_t, ..., P_1^{\tilde{w}^o} \le \alpha_1 \mid E_z^{(t)}, Z_t = z_{\sigma(t)}\} \cdot \mathbb{P}\{Z_t = z_{\sigma(t)} \mid E_z^{(t)}\} \\
&= \sum_{\sigma(t)=1}^{t} \mathbb{E}\big[\mathbb{1}\{P_t^{\tilde{w}^o} \le \alpha_t, ..., P_1^{\tilde{w}^o} \le \alpha_1\} \mid E_z^{(t)}, Z_t = z_{\sigma(t)}\big] \cdot \mathbb{P}\{Z_t = z_{\sigma(t)} \mid E_z^{(t)}\} \\
&= \sum_{\sigma(t)=1}^{t} \mathbb{E}\big[\mathbb{1}\{P_t^{\tilde{w}^o} \le \alpha_t\} \cdot \mathbb{1}\{P_{t-1}^{\tilde{w}^o} \le \alpha_{t-1}, ..., P_1^{\tilde{w}^o} \le \alpha_1\} \mid E_z^{(t)}, Z_t = z_{\sigma(t)}\big] \cdot \mathbb{P}\{Z_t = z_{\sigma(t)} \mid E_z^{(t)}\}
\end{aligned}
$$

Next, observe that conditioning on $E_z^{(t)}$ and $Z_t = z_{\sigma(t)}$ in the expectation term settles the value of $\mathbb{1}\{P_t^{\tilde{w}^o} \le \alpha_t\}$ to be $\mathbb{1}\{p_{\sigma(t)}^{\tilde{w}^o} \le \alpha_t\}$; that is, noting that the score function $\widehat{\mathcal{S}}$ is bijective, $\{E_z^{(t)}, Z_t = z_{\sigma(t)}\} \implies \{E_z^{(t)}, V_t = v_{\sigma(t)}\} \implies P_t^{\tilde{w}^o} = p_{\sigma(t)}^{\tilde{w}^o}$. So, $\mathbb{1}\{P_t^{\tilde{w}^o} \le \alpha_t\} = \mathbb{1}\{p_{\sigma(t)}^{\tilde{w}^o} \le \alpha_t\}$ can be pulled out from the expectation to obtain

$$
\begin{aligned}
&\mathbb{P}\{P_t^{\tilde{w}^o} \le \alpha_t, ..., P_1^{\tilde{w}^o} \le \alpha_1 \mid E_z^{(t)}\} \\
&= \sum_{\sigma(t)=1}^{t} \mathbb{1}\{p_{\sigma(t)}^{\tilde{w}^o} \le \alpha_t\} \cdot \mathbb{E}\big[\mathbb{1}\{P_{t-1}^{\tilde{w}^o} \le \alpha_{t-1}, ..., P_1^{\tilde{w}^o} \le \alpha_1\} \mid E_z^{(t)}, Z_t = z_{\sigma(t)}\big] \cdot \mathbb{P}\{Z_t = z_{\sigma(t)} \mid E_z^{(t)}\}.
\end{aligned}
$$

Now, observe that conditioning on $E_z^{(t)}$ and $Z_t = z_{\sigma(t)}$ implies that $\{Z_1, ..., Z_{t-1}\} = \{z_1, ..., z_t\} \backslash \{z_{\sigma(t)}\}$. That is, observing the bag of $t$ data values (i.e., observing $E_z^{(t)}$) and the value taken on by the $t$-th random variable ($Z_t = z_{\sigma(t)}$) implies that each of $Z_1, ..., Z_{t-1}$ takes a value in $\{z_1, ..., z_t\} \backslash \{z_{\sigma(t)}\}$.

Without loss of generality, we can relabel the indices on the data values as $\{z_1, ..., z_{t-1}\} \leftarrow \{z_1, ..., z_t\} \backslash \{z_{\sigma(t)}\}$, and so we denote this event $\{Z_1, ..., Z_{t-1}\} = \{z_1, ..., z_{t-1}\}$ as $E_z^{(t-1)}$. The WCP $p$-variable $P_{t-1}^{\tilde{w}^o}$ is determined by drawing a sequence $z_{\sigma(1)}, ..., z_{\sigma(t-1)}$ (i.e., determined by a permutation $\sigma : [t-1] \rightarrow [t-1]$) from the bag of data values $\{z_1, ..., z_{t-1}\}$ with probability proportional to $f_{Z_1,...,Z_{t-1}}(z_{\sigma(1)}, ..., z_{\sigma(t-1)})$ and then applying the the nonconformity score function $\mathcal{S}$, and so $E_z^{(t-1)}$ is a "sufficient statistic" for $P_{t-1}^{\tilde{w}^o}$ (for a known $f_{Z_1,...,Z_{t-1}}$). In other words, we can substitute the conditioning on the event $E_Z^{(t)}$ and $Z_t = z_{\sigma(t)}$ with conditioning on $E_Z^{(t-1)}$:

$$\mathbb{P}\{P_t^{\tilde{w}^o} \leq \alpha_t, ..., P_1^{\tilde{w}^o} \leq \alpha_1 \mid E_z^{(t)}\}$$
$$= \sum_{\sigma(t)=1}^{t} \mathbb{1}\{p_{\sigma(t)}^{\tilde{w}^o} \leq \alpha_t\} \cdot \mathbb{P}\{P_{t-1}^{\tilde{w}^o} \leq \alpha_{t-1}, ..., P_1^{\tilde{w}^o} \leq \alpha_1 \mid E_z^{(t-1)}\} \cdot \mathbb{P}\{Z_t = z_{\sigma(t)} \mid E_z^{(t)}\}.$$

Using the inductive assumption and Lemma B.1, this becomes

$$\mathbb{P}\{P_t^{\tilde{w}^o} \leq \alpha_t, ..., P_1^{\tilde{w}^o} \leq \alpha_1 \mid E_z^{(t)}\} = \sum_{\sigma(t)=1}^{t} \mathbb{1}\{p_{\sigma(t)}^{\tilde{w}^o} \leq \alpha_t\} \cdot \alpha_{t-1} \cdots \alpha_1 \cdot \mathbb{P}\{Z_t = z_{\sigma(t)} \mid E_z^{(t)}\}$$
$$= \mathbb{P}\{P_t^{\tilde{w}^o} \leq \alpha_t \mid E_z^{(t)}\} \cdot \alpha_{t-1} \cdots \alpha_1$$
$$= \alpha_t \cdots \alpha_1,$$

which proves Eq. (20). Note that Eq. (20) is a conditional result for any $t = 1, ..., T$; marginalizing over the event $E_z^{(t)}$ and taking $t = T$ implies

$$\mathbb{P}\{P_T^{\tilde{w}^o} \leq \alpha_T, ..., P_1^{\tilde{w}^o} \leq \alpha_1\} = \alpha_T \cdots \alpha_1. \tag{21}$$

We have proven that $P_1^{\tilde{w}^o}, P_2^{\tilde{w}^o}, ..., P_T^{\tilde{w}^o} \overset{iid}{\sim} \mathrm{Unif}[0,1]^T$; this implies the analogous result for the infinite sequence, that is, $P_1^{\tilde{w}^o}, P_2^{\tilde{w}^o}, ... \overset{iid}{\sim} \mathrm{Unif}[0,1]^\infty$ (Shiryaev, 2016; Vovk et al., 2003).

Note that Eq. (21) holds in theory for an arbitrary joint PDF $f$, but it is an abstract statement because it relies on the oracle weights in Eq. (19), which are intractable due to requiring knowledge of $f$ and factorial complexity. Thus, if the oracle weights in Eq. (19) are simplified using some assumptions or approximations on $f$ (e.g., conditional independence or invariance assumptions, density-ratio estimations) that we denote by $\mathcal{H}_0(\hat{f})$, and denoting the resulting approximated weights as $\tilde{w}$, then the $P_t^{\tilde{w}}$ are IID uniformly distributed, assuming $\mathcal{H}_0(\hat{f})$:

$$\mathbb{P}_{\mathcal{H}_0(\hat{f})}\{P_T^{\tilde{w}} \leq \alpha_T, ..., P_1^{\tilde{w}} \leq \alpha_1\} = \alpha_T \cdots \alpha_1. \tag{22}$$

Eq. (22) implies that $P_1^{\tilde{w}}, P_2^{\tilde{w}}, ..., P_T^{\tilde{w}} \mid \mathcal{H}_0(\hat{f}) \overset{iid}{\sim} \mathrm{Unif}[0,1]^T$. Similarly as before, this result for a fixed number of points $T$ implies the corresponding result for infinite sequences (Shiryaev, 2016; Vovk et al., 2003), that is, $P_1^{\tilde{w}}, P_2^{\tilde{w}}, ... \mid \mathcal{H}_0(\hat{f}) \overset{iid}{\sim} \mathrm{Unif}[0,1]^\infty$.

### B.2. Proofs for Proposition 3.2 (WCTM Anytime False-Alarm Control) and Proposition 3.3 (Average-Run-Length Control for WCTM-based Shiryaev-Roberts Procedure)

The proofs for Proposition 3.2 and Proposition 3.3 follow from Theorem 3.1 and Vovk (2021)'s results for standard conformal test martingales. By Theorem 3.1, weighted conformal $p$-values have IID uniform distribution on [0,1] premised on some assumptions $\mathcal{H}_0(\hat{f})$. Thus, a sequence of weighted conformal $p$-values is a sequence of IID random variables drawn from Unif[0,1], assuming that the null hypothesis $\mathcal{H}_0(\hat{f})$ is true. Because the construction of a weighted conformal test martingale (e.g., Eq. (7)) is only allowed to depend on the data via the sequence of weighted conformal $p$-values, we can draw from Vovk (2021)'s results that construct betting martingales that only assume a sequence of IID uniform [0,1] random variables. That is, by construction, weighted conformal test martingales are a betting martingale (Vovk, 2021) whose validity is premised on the assumptions $\mathcal{H}_0(\hat{f})$ (used to compute the weights). Then, Proposition 3.2 follows by Ville's inequality (Ville, 1939) and Proposition 3.3 follows from Proposition 4.2 in Vovk (2021).

# C. Deriving Specific WCTM Algorithms for Main Practical Testing Objective: Testing for {Concept Shift or Unanticipated Covariate Shift}

In this section we describe the derivation for the specific WCTM algorithms implemented in this paper for our primary testing objective, of detecting either concept shift (in $Y \mid X$) or unanticipated covariate shift (extreme shift in $X$). Because a WCTM is constructed on top of a corresponding weighted CP method by a betting process on the associated sequence of weighted CP $p$-values, this will initially parallel the procedure described in Prinster et al. (2024) for deriving weighted CP methods. The resulting weighted CP method (that the WCTM algorthims are built on top of) will be similar to that used in Tibshirani et al. (2019), except whereas that paper focuses on CP for a shift between two batches of data, we focus on an online monitoring setting where a shift may occur at an unknown post-deployment time and density-ratio weights may be estimated online.

*Step 1: List assumptions/null hypothesis of interest*

We begin by restating the null hypothesis (i.e., the nonparametric assumptions on what aspects of the data distribution are expected to stay invariant versus change) that is of primary practical interest in this paper. That is, as in Eq. (13) in the main paper, our null hypothesis $\mathcal{H}_0^{(\mathrm{cs})}$ assumes that the conditional label distribution $Y \mid X$ remains invariant and that the marginal $X$ distribution can shift, but that at some post-deployment time $t_{\mathrm{ad}}$ (the time at which to begin adaptation to covariate shift), the covariates $X$ shift in distribution in a manner described by the (estimated) density-ratio function $\widehat{w}(x)$. That is, we want to sequentially test the null hypothesis

$$\mathcal{H}_0^{(\mathrm{cs})} := \begin{cases} (X_i, Y_i) & \overset{iid}{\sim} \mathbf{F}_{\mathbf{Y}|\mathbf{X}} \times \mathbf{F}_{\mathbf{X}}, & i \in [n + t_{\mathrm{ad}} - 1] \\ (X_{n+t}, Y_{n+t}) & \overset{iid}{\sim} \mathbf{F}_{\mathbf{Y}|\mathbf{X}} \times \mathbf{G}_{\mathbf{X}}^{(\widehat{\mathbf{w}})}, & t \geq t_{\mathrm{ad}}, \end{cases} \tag{23}$$

where boldface is used to denote a set of distributions, and in particular $\mathbf{G}_{\mathbf{X}}^{(\widehat{\mathbf{w}})}$ denotes the set of distributions for $X$ with density-ratio function $\widehat{w}(x)$ (with respect to the true but unknown source covariate distribution $F_X \in \mathbf{F}_{\mathbf{X}}$):

$$\mathbf{G}_{\mathbf{X}}^{(\widehat{\mathbf{w}})} := \left\{ G_X \ : \ \mathrm{d}G_X(x)/\mathrm{d}F_X(x) = \widehat{w}(x) \quad \text{if} \quad \mathrm{d}F_X(x) > 0 \right\}.$$

The density-ratio function $\widehat{w}(x)$ can be estimated with offline data and/or continually updated as each test point is observed.

*Step 2: Factorize joint density using null hypothesis assumptions*

We now use our null hypothesis assumptions $\mathcal{H}_0^{(\mathrm{cs})}$ to factorize the joint probability density function. See Prinster et al. (2024)'s Appendix B.1 for a more detailed derivation for a more general setting.

$$f(z_1, ..., z_{n+t}) = f_X(x_1) \cdots f_X(x_{n+t_{\mathrm{ad}}-1}) \cdot g_X(x_{n+t_{\mathrm{ad}}}) \cdots g_X(x_{n+t}) \cdot \prod_{j=1}^{n+t} \left[ f_{Y|X}(y_j|x_j) \right]$$

$$= \prod_{j=1}^{n+t_{\mathrm{ad}}-1} \left[ f_X(x_j) \right] \cdot \prod_{j=n+t_{\mathrm{ad}}}^{n+t} \left[ g_X(x_j) \right] \cdot \prod_{j=1}^{n+t} \left[ f_{Y|X}(y_j|x_j) \right],$$

multiplying by $1 = \frac{\prod_{j=n+t_{\mathrm{ad}}}^{n+t} f_X(x_j)}{\prod_{j=n+t_{\mathrm{ad}}}^{n+t} f_X(x_j)}$, we have

$$f(z_1, ..., z_{n+t}) = \prod_{j=1}^{n+t} \left[ f_X(x_j) \right] \cdot \prod_{j=n+t_{\mathrm{ad}}}^{n+t} \left[ \frac{g_X(x_j)}{f_X(x_j)} \right] \cdot \prod_{j=1}^{n+t} \left[ f_{Y|X}(y_j|x_j) \right],$$

and approximating the density-ratio function $\frac{g_X(x_j)}{f_X(x_j)}$ with our estimate $\widehat{w}(x_j)$, this becomes

$$f(z_1, ..., z_{n+t}) = \prod_{j=n+t_{\text{ad}}}^{n+t} \widehat{w}(x_j) \cdot \prod_{j=1}^{n+t} \left[ f_{Y|X}(y_j|x_j) \cdot f_X(x_j) \right]$$

$$= \underbrace{\prod_{j=n+t_{\text{ad}}}^{n+t} \widehat{w}(x_j)}_{\text{Time-dependent factors}} \cdot \underbrace{\prod_{j=1}^{n+t} f_Z(z_j)}_{\text{Time-invariant factor}} . \tag{24}$$

*Step 3: Compute or estimate weights and weighted-conformal p-values*

Recalling the definition of the CP weights from the main paper (Eq. (10)) and plugging in the factorization from Eq. (24):

$$\mathbb{P}\{Z_{n+t} = z_i \mid E_z^{(t)}\} = \frac{\sum_{\sigma:\sigma(n+t)=i} f(z_{\sigma(1)}, ..., z_{\sigma(n+t)})}{\sum_{\sigma} f(z_{\sigma(1)}, ..., z_{\sigma(n+t)})}$$

$$= \frac{\sum_{\sigma:\sigma(n+t)=i} \prod_{j=n+t_{\text{ad}}}^{n+t} \widehat{w}(x_{\sigma(j)}) \cdot \prod_{j=1}^{n+t} f_Z(z_{\sigma(j)})}{\sum_{\sigma} \prod_{j=n+t_{\text{ad}}}^{n+t} \widehat{w}(x_{\sigma(j)}) \cdot \prod_{j=1}^{n+t} f_Z(z_{\sigma(j)})}$$

$$= \frac{\sum_{\sigma:\sigma(n+t)=i} \prod_{j=n+t_{\text{ad}}}^{n+t} \widehat{w}(x_{\sigma(j)})}{\sum_{\sigma} \prod_{j=n+t_{\text{ad}}}^{n+t} \widehat{w}(x_{\sigma(j)})}, \tag{25}$$

where the last line follows because the factor $\prod_{j=1}^{n+t} f_Z(z_{\sigma(j)})$ is invariant to permutations and thus cancels in the ratio. However, the computational complexity for computing Eq. (25) is still $n + t$ choose $t_{\text{ad}}$ (i.e., requiring $\frac{(n+t)!}{t_{\text{ad}}!(n+t-t_{\text{ad}})!}$ computations), so to enable tractabile estimation, we focus only on permuting the test point value with values assumed to be from the source distribution (prior to the adaptation beginning); that is, we the sums to permutations $\sigma' : [n+t] \to [n+t]$ such that $\sigma'(j) = j$ for all $j \in \{n + t_{\text{ad}}, ..., n + t - 1\}$:

$$\mathbb{P}\{Z_{n+t} = z_i \mid E_z^{(t)}\} \approx \tilde{w}_i^{(t)} := \frac{\sum_{\sigma':\sigma'(n+t)=i} \widehat{w}(x_{\sigma'(n+t)})}{\sum_{\sigma'} \widehat{w}(x_{\sigma'(n+t)})}$$

$$= \frac{\widehat{w}(x_i)}{\sum_{i \in [n+t_{\text{ad}}-1] \cup \{n+t\}} \widehat{w}(x_i)}. \tag{26}$$

Then, the corresponding weighted-conformal p-value is

$$p_{n+t}^{\tilde{w}} = \sum_{i \in [n+t_{\text{ad}}-1] \cup \{n+t\}} \tilde{w}_i^{(t)} \left[ \mathbb{1}\{v_i > v_{n+t}\} + u_{n+t} \mathbb{1}\{v_i = v_{n+t}\} \right]. \tag{27}$$

Due to the modification made in Eq. (26), Theorem 3.1 only directly applies for the covariate shift null hypothesis (i.e., for testing $\mathcal{H}_0(\hat{f}) = \mathcal{H}_0^{(\text{cs})}$ as in Eq. (23)) for the timepoint $t = t_{\text{ad}}$. This is due to that, when a weighted-conformal p-value is not computed fully online,[9] the proof-by-induction approach in Section B.1 may not apply; in other words, for two test points $n + t$ and $n + t'$, their p-values $p_{n+t}^{\tilde{w}}$ and $p_{n+t'}^{\tilde{w}}$ may not be statistically independent due to their common dependency on the calibration data $[n + t_{\text{ad}} - 1]$ in Eq. (27). This gap can be resolved to enable testing of Eq. (23) if the calibration data $[n + t_{\text{ad}} - 1]$ are resampled for each test point, which restores independence trivially (due to each calibration set being entirely new). Alternatively, we can modify Eq. (23) to also assume that we have enough initial source data (i.e., $n$ is large enough) so that the calibration scores' empirical CDF is equal to the true source score distribution, that is, $\widehat{F}_V^{[n+t_{\text{ad}}-1]} = F_V$,

---

[9]That is, not computed relative to all the full observed data sequence with indices $[n + t]$, but only relative to a strict subset of $[n + t]$; for example, Eq. (27) computes the p-value relative to the data with indices $[n + t_{\text{ad}} - 1] \cup \{n + t\}$, where $t_{\text{ad}} < t$.

which is of course true in the limit $n \to \infty$. Accordingly, the $p$-value in Eq. (27) is effectively testing the modified null hypothesis of $\mathcal{H}_0^{(\text{cs})}$ *and* $\widehat{F}_V^{[n+t_{\text{ad}}-1]} = F_V$:

$$\mathcal{H}_0^{(\text{cs})} \wedge (\widehat{F}_V^{[n+t_{\text{ad}}-1]} = F_V) = \begin{cases} (X_i, Y_i) & \overset{iid}{\sim} \mathbf{F}_{\mathbf{Y}|\mathbf{X}} \times \mathbf{F}_{\mathbf{X}}, & i \in [n + t_{\text{ad}} - 1] \\ (X_{n+t}, Y_{n+t}) \overset{iid}{\sim} \mathbf{F}_{\mathbf{Y}|\mathbf{X}} \times \mathbf{G}_{\mathbf{X}}^{(\widehat{\mathbf{w}})}, & t \geq t_{\text{ad}} \\ \widehat{F}_V^{[n+t_{\text{ad}}-1]} = F_V. \end{cases} \quad (28)$$

In practice, Eq. (28) can roughly be thought of as also testing whether the weighted CP calibration set is "large enough" for CP coverage to hold precisely *conditional on* the calibration data. That is, letting $Z_{\text{cal}}$ denote the event that $\{Z_1, ..., Z_{n+t_{\text{ad}}-1}\} = \{z_1, ..., z_{n+t_{\text{ad}}-1}\}$, taking Eq. (28) as given would imply that, for all $\alpha \in (0, 1)$,

$$\mathbb{P}\{P_{n+t}^{\tilde{w}} \leq \alpha \mid Z_{\text{cal}}\} = \alpha \quad \Longleftrightarrow \quad \mathbb{P}\{Y_{n+t} \in \widehat{C}_{[n+t_{\text{ad}}-1],\alpha}(X_{n+t}) \mid Z_{\text{cal}}\} = 1 - \alpha, \quad (29)$$

where $P_{n+t}^{\tilde{w}}$ is the random variable taking values in Eq. (23) and the probability is only over the draw of the test point $Z_{n+t}$. In other words, Eq. (28) $\implies$ Eq. (29), so via the contrapositive, violations of calibration-set-conditional coverage (Eq. (29)) would imply violations of $\mathcal{H}_0^{(\text{cs})} \wedge (\widehat{F}_V^{[n+t_{\text{ad}}-1]} = F_V)$ (Eq. (28)).

*Step 4: Construct WCTM by a betting process on the weighted-conformal $p$-values*

Lastly, the practical WCTM algorithms implemented in this paper are constructed on top of a sequence of weighted-conformal $p$-values $p_{n+1}^{\tilde{w}}, p_{n+2}^{\tilde{w}}, ..., p_{n+t}^{\tilde{w}}, ...$, as defined in Eq. (27). The resulting WCTM thus achieves the anytime-valid false alarm guarantee in Proposition 3.2 for the null hypothesis $\mathcal{H}_0(\hat{f}) = \mathcal{H}_0^{(\text{cs})} \wedge (\widehat{F}_V^{[n+t_{\text{ad}}-1]} = F_V)$, as defined in Eq. (28) (and analogously for the scheduled, Shiryaev-Roberts monitoring procedure based on WCTMs).

# D. Additional Experimental Results

## D.1. Root-Cause Analysis with WCTMs

## D.2. What Makes a Covariate Shift Mild, Moderate, or Extreme?

While the distinction between mild, moderate, and extreme covariate shifts can be gradual, problem-specific, and sometimes user-driven, these disctinctions are not arbitrary, and WCTMs arguably even provide an approach to such delineation with statistical guarantees. Intuitively, benign shifts can be considered those where the "safety" of the CP coverage is maintained nontrivially (i.e., without the prediction set covering the whole label space). This intuition corresponds to the WCTMs' null hypothesis (and, harmful shifts violate it), as follows:

- **Benign:** The betting martingale's null hypothesis is that the WCP $p$-values are IID Unif$[0, 1]$ (Appendix B.2); this null implies that coverage is satisfied (exactly) for all $\alpha \in (0, 1)$ (ie, the martingale's null $\implies$ intuitive definition of "benign" regarding coverage validity).

- **Harmful:** (Contrapositive of the above.) If coverage is *not* satisfied (exactly) for some $\alpha \in (0, 1)$, then the $p_t^{\tilde{w}}$ are not IID Unif$[0, 1]$ (ie, violation of coverage validity $\implies$ violation of martingale's null, thus possibility for detection). Larger violations are easier to detect, and thus more likely to quickly raise an alarm. Note that this can be due to under-coverage (safety violation) or over-coverage (uninformative prediction sets); we further penalize trivial overcoverage—i.e., when $\widehat{C}(X_{n+t}) = \mathcal{Y}$—by using anticonservative WCP $p$-values whenever this occurs. (See pseudocode in Appendix F.)

- **Medium:** A shift may initially be "harmful" as described above, due to density-ratio estimator having insufficient data, but later become "benign" once enough data has been collected.

Figure 5 provides an ablation study illustrating synthetic data example of WATCH performance for different magnitudes of covariate shift (in the input $X$ distribution). Each row corresponds to a specific magnitude of covariate shift and illustrates WATCH's response regarding coverage (prediction safety), interval widths (prediction informativeness), and WCTMs (monitoring criteria for alarms). The post-changepoint test points are sampled from the full source distribution with probabilities proportional to $exp(|x - 18| * \lambda)$; larger values of $\lambda$ thus correspond to more severe covariate shift toward extreme (and particularly toward large) values of the input $X$. Experiments are averaged over 20 seeds.

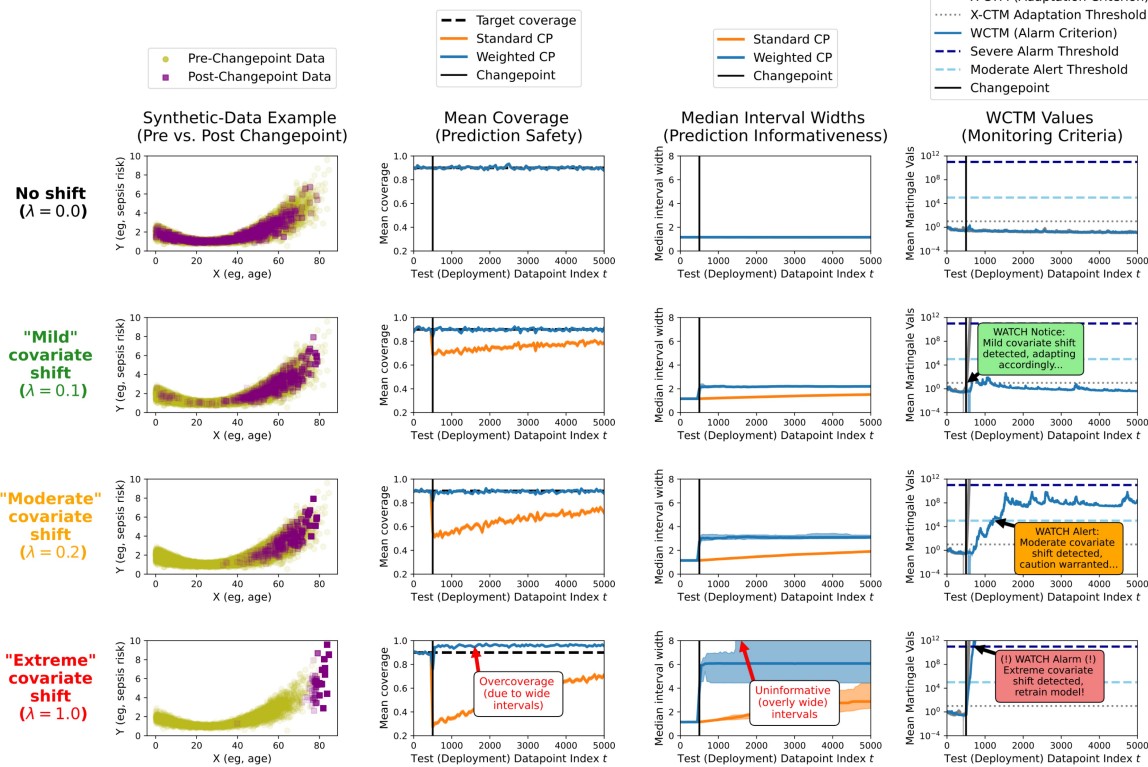

Figure 5: Ablation study illustrating synthetic data example of WATCH performance for different magnitudes of covariate shift (in the input $X$ distribution).

## D.3. Wall-Clock Runtimes

Table 2 presents wall-clock runtimes averaged over 20 random seeds for the same experimental settings as reported in the average detection delay (ADD) experiments in Table 1, except with the true changepoint occurring 200 datapoints after deployment (whereas the results for Table 1 are for 100 samples after deployment). Across all datasets and for both the anytime-valid and the scheduled monitoring criteria, the CTM and WCTM-based methods have faster empirical runtime than baselines. The starkest difference is between the Shiyaev-Roberts (W)CTM methods and the online changepoint detection method from Podkopaev and Ramdas (2021b), due to the former's runtime having complexity $O(t)$ and the latter's complexity being $O(t^2)$, where $t$ is the number of post-deployment timesteps.

Table 2: Wall-clock runtimes in seconds, averaged over 20 random seeds for each monitoring method and dataset (error bars are standard errors). Anytime-valid monitoring methods are WCTMs (proposed), CTMs (Vovk, 2021), and sequential testing from Podkopaev and Ramdas (2021b) (PR-ST); multistage monitoring methods are Shiryaev-Roberts (SR) procedure applied to WCTMs (proposed), SR applied to CTMs (Vovk, 2021), changepoint detection on the miscoverage risk from Podkopaev and Ramdas (2021b) in both online (PR-CD-online) and minibatched (PR-CD-50) variants. Results corresponding to our method are in blue. The best runtimes are **bolded**.

| Dataset ($\downarrow$) | Monitoring Criterion ($\rightarrow$) | Anytime-Valid Criterion | | Scheduled, Multistage Criterion | | |
|---|---|---|---|---|---|---|
| | Monitoring Method ($\rightarrow$) | (W)CTM | PR-ST | SR via (W)CTM | PR-CD-online | PR-CD-50 |
| | CP Type ($\downarrow$) | Time ($\pm$ SE) | Time ($\pm$ SE) | Time ($\pm$ SE) | Time ($\pm$ SE) | Time ($\pm$ SE) |
| **MEPS** | Weighted | **0.38** ($\pm$0.01) | 7.65 ($\pm$1.29) | **0.73** ($\pm$0.05) | 81.75 ($\pm$10.17) | 0.15 ($\pm$0.01) |
| | Standard | **0.38** ($\pm$0.01) | 5.55 ($\pm$1.01) | **0.73** ($\pm$0.05) | 81.97 ($\pm$1.01) | 0.15 ($\pm$0.01) |
| **Bike Sharing** | Weighted | **0.44** ($\pm$0.02) | 10.26 ($\pm$1.72) | **0.87** ($\pm$0.08) | 138.11 ($\pm$16.36) | 0.14 ($\pm$0.01) |
| | Standard | **0.43** ($\pm$0.02) | 8.45 ($\pm$1.27) | **0.88** ($\pm$0.09) | 135.94 ($\pm$16.92) | 0.14 ($\pm$0.01) |
| **Superconduct** | Weighted | **0.43** ($\pm$0.02) | 9.93 ($\pm$1.32) | **0.88** ($\pm$0.09) | 215.72 ($\pm$20.63) | 0.20 ($\pm$0.01) |
| | Standard | **0.42** ($\pm$0.02) | 9.94 ($\pm$1.03) | **0.86** ($\pm$0.09) | 213.72 ($\pm$23.82) | 0.18 ($\pm$0.01) |

### D.4. Ablation Experiments for Density-Ratio Weight Estimation

Another factor determining whether a covariate shift can be considered benign or harmful to deployment is whether a deployed density-ratio estimator is well-specified and thus able to approximate the shift. Figure 6 provides selected synthetic-data example where logistic regression is a misspecified probabilistic classifier for distinguishing between pre- and post-changepoint data, but where a neural network (MLP) is able to accurately discriminate between the same pre- and post-changepoint data. That is, in this example the pre- and post-changepoint data are not linearly separable in the input $X$ domain, so logistic regression is not able to reliably discriminate, and thereby it is unable to reliably estimate density-ratio weights via probabilistic classification. The result is that the changepoint causes a large increase in coverage, despite some adaptation (decreasing interval widths); the estimator's misspecification thus causes WCTMs to raise an alarm, indicating that the covariate shift cannot be adapted to by the estimator. In contrast, the MLP estimator is able to appropriately adapt by maintaing target coverage, improving interval sharpness, and avoiding unnecessary alarms.

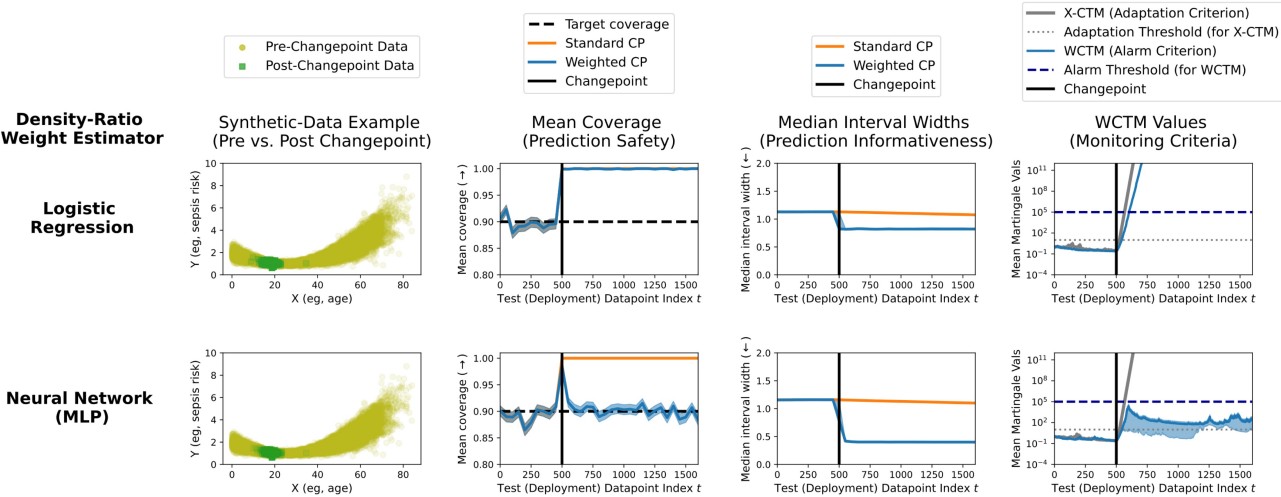

Figure 6: Ablation study on density-ratio estimator for synthetic data.

### D.5. Ablation Experiments for Betting Function

The primary role of the betting function in (W)CTMs and testing-by-betting more broadly is for quickly rejecting the null hypothesis (i.e., raising an alarm) when it is violated. Figure 7 provides ablation experiment on the betting function used for X-CTMs and WCTMs, on three settings of the synthetic-data example. The "Composite" Jumper betting function is the betting function used in all other experiments, and it is an average of Simple Jumper betting functions over "jumping parameters" $J \in [0.0001, 0.001, 0.01, 0.1, 1]$; here, we set the Simple Jumper baseline to have $J = 0.01$. See Vovk et al. (2021) for pseudocode and exposition of the Simple Jumper algorithm. $J = 1$ means conservatively spreading bets across all options to avoid cumulative losses, while smaller $J$ encourages "doubling down" on bets that were previously successful. The CTMs with Composite betting are thus lower bounded at $M_t = 0.2$, whereas those with Simple betting continually decrease, resulting in slightly delayed detection speed relative to Composite.

### D.6. Further Experiments on Corrupted Image Data

Here we provide additional experiments on the corrupted image data. Our evaluation metrics include the average detection delay (ADD), the average number of unnecessary alarms, and the average number of missed alarms. First, WCTM exhibits shorter detection delays compared to CTM. Given that CTM often doesn't trigger alarms in the face of severe corruption, which significantly increases its overall ADD. Additionally, as the corruption level intensifies, WCTM detects shifts more quickly, whereas CTM's detection speed shows minimal change—consistent with our earlier visualizations. To evaluate unnecessary alarms, we again create a mild, benign shift in the target distribution by mixing clean samples into MNIST and level-1 CIFAR-10 corruption data, while treating higher-level corruptions as harmful. We find that WCTM's unnecessary alarm rate is roughly *one-third* that of CTM. Moreover, WCTM also exhibits fewer missed alarms, especially under more severe corruptions. These findings highlight the flexibility of our framework, which not only avoids unnecessary alarms but

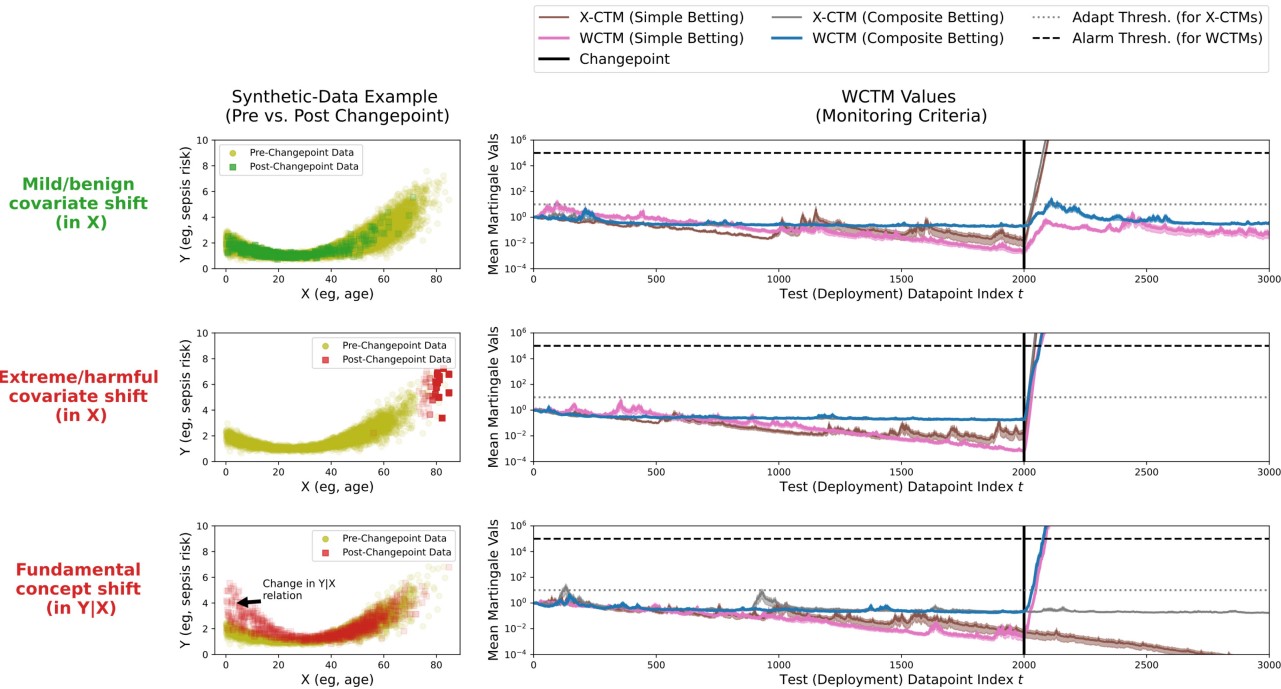

Figure 7: Ablation study on betting function.

Table 3: WCTM achieves a lower Average Detection Delay (ADD), especially under severe image corruptions, and also exhibits fewer false alarms and missed alarms. Results are averaged over 10 random seeds and all corruption types.

|  |  | MNIST-C | CIFAR10-C L1 | CIFAR10-C L3 | CIFAR10-C L5 |
|---|---|---|---|---|---|
| **ADD** | WCTM | **156.4** | 188.0 | **163.6** | **129.7** |
|  | CTM | 285.3 | **176.5** | 175.2 | 170.3 |
| **Unnecessary Alarm** | WCTM | **7.3** | **12.2** | – | – |
|  | CTM | 25.4 | 34.3 | – | – |
| **Missed Alarm** | WCTM | 2.9 | 6.6 | **3.8** | **0.9** |
|  | CTM | **2.4** | **5.9** | 4.6 | 1.2 |

also better detects harmful shifts.

Whereas Table 3 provided metrics averaged over all corruption types, Figure 8 provides specific example results on each corruption type separately, focusing on harmful shifts with high severity of corruption. These plots illustrate how WCTMs quickly react to harmful shifts.

# E. Experiment Details

### E.1. Datasets

The evaluation datasets include both real-world tabular and image datasets. The tabular datasets are for regression tasks (where conformal methods used the absolute value residual nonconformity score $\widehat{S}(x, y) = |y - \widehat{\mu}(x)|$), and the datasest span various sizes and dimensionalities: the Medical Expenditure Panel Survey (MEPS) dataset (33005 samples, 107 features) (Cohen et al., 2009), the UCI Superconductivity dataset (21263 samples, 81 features) (Hamidieh, 2018), and the UCI bike sharing dataset (17379 samples, 12 features) (Fanaee-T, 2013). The image datasets were for classification tasks (where conformal methods used the one-minus-softmax score $\widehat{S}(x, y) = 1 - \widehat{p}(y_i \mid x_i)$) were the MNIST-corruption (Mu and Gilmer, 2019) (60000 clean samples, 10000 corrupted samples) and CIFAR-10-corruption (Hendrycks and Dietterich, 2019) (50000 clean samples, 10000 corrupted samples), which are standard benchmarks for assessing distribution shifts.

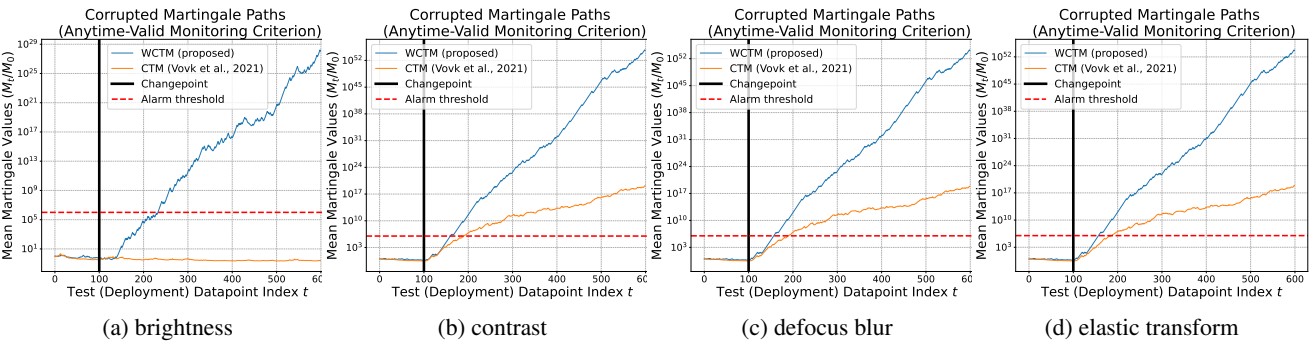

(a) brightness      (b) contrast      (c) defocus blur      (d) elastic transform

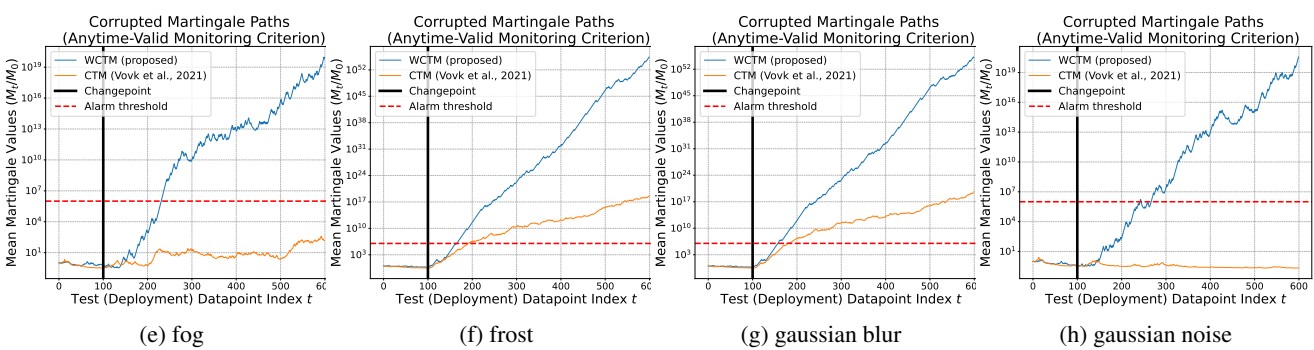

(e) fog      (f) frost      (g) gaussian blur      (h) gaussian noise

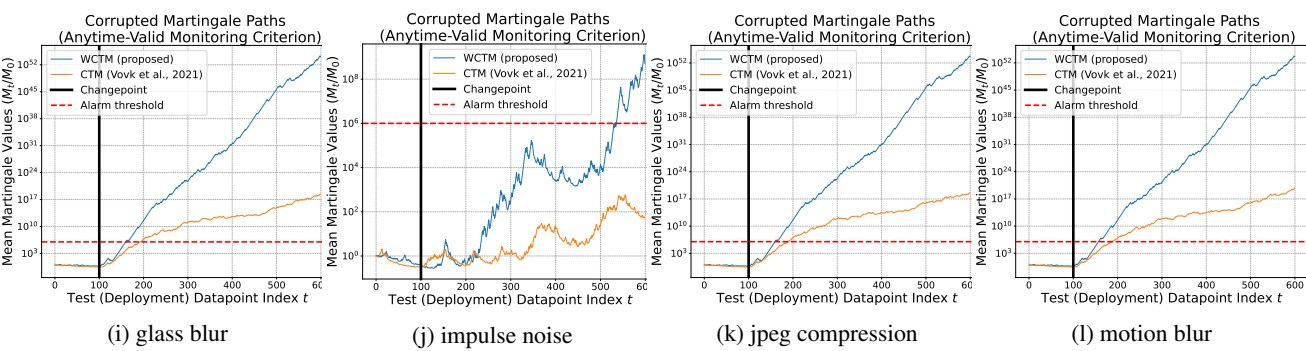

(i) glass blur      (j) impulse noise      (k) jpeg compression      (l) motion blur

Figure 8: Results on CIFAR-10 with various corruption types, all at the highest severity level. WCTM reacts more quickly than the standard CTM under these conditions. Moreover, with several types of corruption, the standard CTM does not raise any alarms at all.

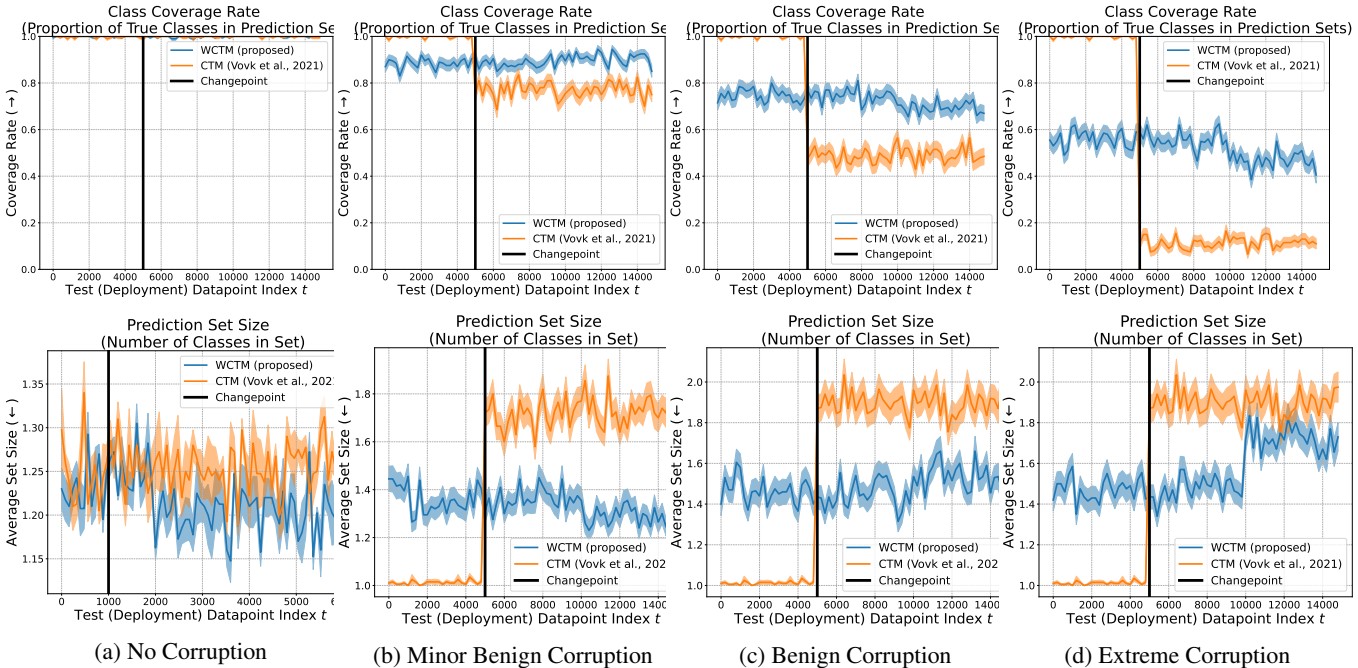

Figure 9: The results supplement Figure 3 in the main paper. They demonstrate the coverage rate and prediction set size under four different corruption scenarios. In the multi-class classification setting, we adopt metrics different from those used in the regression experiments and follow Romano et al. (2020) to measure the prediction set size and coverage rate (defined as the proportion of true classes in the specified range, NOT conformal coverage) as principled risk metrics for distinguishing benign from harmful shifts. We increased the size of the validation set and the number of samples visualized to yield more robust performance and provide a clearer view of the trajectory; the results are averaged over a window size of 200, while all other configurations remain unchanged from the original setting. As discussed in the paper, we mixed test samples (target corrupted) with validation samples (source clean) to improve the estimation of weights for CTMs. So under corrupted scenarios, the "starting points" before the change points for WCTM and CTM differ, as the mixture allows the validation set to contain corrupted data; however, this difference is not clearly reflected in the martingale paths. Overall, the models initially exhibit relatively high classification performance under the clean setting, while CTM rapidly declines to a lower performance level under all corruption conditions. Although WCTM adapts to changes in benign scenarios, it eventually demonstrates severe metric changes under extreme shifts as well, which corresponds to the results in Figure 3.

## E.2. Simulating Shifts in Tabular Data

To evaluate the online adaptation performance of our proposed WCTMs on the tabular, regression-task datasets, we simulated mild or benign covariate shifts by exponentially tilting (i.e., up-sampling) the test samples from the full dataset based on selected covariates. On the MEPS healthcare dataset, $h$ was selected to simulate a demographic shift towards younger, higher-educated patients (expected to be a benign shift due to youth tending to correlate with fewer, less variable health issues). On the bike-sharing dataset, $h$ simulated a shift in weather patterns to colder, windier days; meanwhile, a more complex shift was simulated on the superconductivity dataset by sampling proportional to the projection onto the first principal component representation of the training data. Harmful concept shifts in tabular data were simulated by biasing the label values as a function of selected input covariates for each dataset.

## E.3. Further Details of Image Classification Experiments

We provide further discussions on the image classification experiments from Section 4.2. Figure 9 demonstrates the coverage rates and set sizes of different corruption levels as discussed in the main paper. Details of model architectures and training configurations can be found in Table 4 - Table 7.

Table 4: MNISTDiscriminator Model Details

| Component | Details |
|---|---|
| Model Name | MNISTDiscriminator |
| Purpose | Binary classifier to distinguish between source (uncorrupted) and target (corrupted) MNIST data |
| Architecture Type | Convolutional Neural Network (CNN) with 2 conv blocks + FC layers |
| Input Shape | (Batch_size, 1, 28, 28) - Grayscale MNIST images |
| Output Shape | (Batch_size, 2) - Binary classification logits |
| Total Parameters | $\approx$1.3M parameters |
| Layers | **First Conv Block:** `Conv2d(1→32, 3×3)` → `BatchNorm` → `ReLU` → `MaxPool(2×2)` → `Dropout` 
 **Second Conv Block:** `Conv2d(32→64, 3×3)` → `BatchNorm` → `ReLU` → `MaxPool(2×2)` → `Dropout` 
 **Fully Connected:** `Linear(64×7×7→128)` → `BatchNorm` → `ReLU` → `Dropout` 
 **Output:** `Linear(128→2)` |
| Regularization | `Dropout` (rate=0.3), `BatchNormalization` |
| Calibration Method | Temperature scaling (initial temp=1.5) |
| Activation Functions | `ReLU` |
| Loss Function | Cross-entropy (implicit in the code) |
| Training Epochs | 30 |
| Batch Size | 64 |
| Learning Rate | 0.001 |
| Optimizer | `Adam` |
| Special Features | Temperature scaling parameter for improved probability calibration |
| Usage Context | Used for estimating likelihood ratios in weighted conformal prediction |

---

**Algorithm 1** Calculate weighted conformal prediction set for covariate shift (Tibshirani et al., 2019).

---

**Input:** Calibration data $Z_{1:n} := \{(X_i, Y_i)\}_{i=1}^n$; Test point input $X_{n+1}$; Score function $\hat{s} : (\mathcal{X} \times \mathcal{Y})^* \to (\mathbb{R}^+)^*$; Density-ratio weight function $\hat{w} : \mathcal{X}^* \to [0,1]^*$; Significance level (target miscoverage) $\alpha$.

**Output:** Conformal prediction set $\widehat{C}_{n,\alpha}(X_{n+1}) \subseteq \mathcal{Y}$.

Calculate $n$ nonconformity scores and $n+1$ density-ratio weights:
$v_1, ..., v_n = \hat{s}(Z_1, ..., Z_n)$
$\tilde{w}_1, ..., \tilde{w}_{n+1} = \hat{w}(X_1, ..., X_{n+1})$

Calculate $1 - \alpha$ quantile over weighted score distribution (with conservative adjustment $v_{n+1} = \infty$, as $Y_{n+1}$ is unknown):
$q_{1-\alpha} = \widehat{Q}_{1-\alpha}\big(\sum_{i=1}^n \tilde{w}_i \cdot \delta_{v_i} + \tilde{w}_{n+1} \cdot \delta_\infty\big)$ {Note: $\tilde{w}_{n+1} \geq \alpha \implies q_{1-\alpha} = \infty$.}

Calculate weighted conformal prediction set:
$\widehat{C}_{n,\alpha}(X_{n+1}) = \big\{y \in \mathcal{Y} : \hat{s}(X_{n+1}, y) \leq q_{1-\alpha}\big\}$ {Note: $q_{1-\alpha} = \infty \implies \widehat{C}_{n,\alpha}(X_{n+1}) = \mathcal{Y}$.}

**Return:** $\widehat{C}_{n,\alpha}(X_{n+1})$

---

# F. Algorithms

Only limited algorithm pseudocode is provided at this time, focused on algorithms that explain how we penalized noninformative conformal prediction intervals (i.e., with anticonservative $p$-values). We aim to update the arXiv version of this paper with more comprehensive pseudocode.

Table 5: CIFAR10 Discriminator Model Details

| Component | Details |
|---|---|
| Model Name | CIFAR10Discriminator |
| Purpose | Binary classifier to distinguish between source (uncorrupted) and target (corrupted) CIFAR-10 data |
| Architecture Type | Convolutional Neural Network (CNN) with 3 conv blocks + FC layers |
| Input Shape | (Batch_size, 3, 32, 32) - RGB CIFAR-10 images |
| Output Shape | (Batch_size, 2) - Binary classification logits |
| Total Parameters | $\approx$4.8M parameters |
| Layers | First Conv Block: `Conv2d(3→64, 3×3)` → `BatchNorm` → `ReLU` → `MaxPool(2×2)` → `Dropout` 
 Second Conv Block: `Conv2d(64→128, 3×3)` → `BatchNorm` → `ReLU` → `MaxPool(2×2)` → `Dropout` 
 Third Conv Block: `Conv2d(128→256, 3×3)` → `BatchNorm` → `ReLU` → `MaxPool(2×2)` → `Dropout` 
 First FC Layer: `Linear(256×4×4→512)` → `BatchNorm` → `ReLU` → `Dropout` 
 Second FC Layer: `Linear(512→128)` → `BatchNorm` → `ReLU` → `Dropout` 
 Output: `Linear(128→2)` |
| Regularization | `Dropout` (rate=0.3), `BatchNormalization` at each layer |
| Calibration Method | Temperature scaling (initial temp=1.5) |
| Activation Functions | `ReLU` throughout the network |
| Loss Function | Cross-entropy |
| Training Epochs | 30 |
| Batch Size | 64 |
| Learning Rate | 0.001 |
| Optimizer | `Adam` |
| Special Features | Temperature scaling parameter for improved probability calibration |
| Usage Context | Used for estimating likelihood ratios in weighted conformal prediction |

---

**Algorithm 2** Calculate weighted conformal $p$-value that penalizes noninformativeness.

---

**Input:** Calibration scores $V_{1:n} := \{V_1, ..., V_n\}_{i=1}^n$; Test point score $V_{n+1}$; Weight vector $\widetilde{w} := (\widetilde{w}_1, ..., \widetilde{w}_{n+1})$; Significance level (target miscoverage) $\alpha$.

**Output:** Weighted conformal $p$-value $p_{n+1}^{\tilde{w}}$.

**if** $\widetilde{w}_{n+1} < \alpha$ **then**

    `Calculate exact weighted conformal `$p$`-value (Eq. (9)):`

    Sample $u_{n+1} \overset{iid}{\sim} \text{Unif}[0,1]$

    $p_{n+1}^{\tilde{w}} = \sum_{i=1}^{n+1} \tilde{w}_i \left[ \mathbb{1}\{v_i > v_{n+1}\} + u_{n+1}\mathbb{1}\{v_i = v_{n+1}\} \right]$.

**else**

    {Note: This "else" condition implies the corresponding CP set for the test point was noninformative, i.e., $\widehat{C}_{n,\alpha}(X_{n+1}) = \mathcal{Y}$; thus, penalize noninformativeness by using derandomized, anticonservative $p$-value, equivalent to setting $u_{n+1} = 0$. These will cause WCTM to raise an alarm more quickly, as they intentionally break IID uniformity based on the magnitude of $\tilde{w}_{n+1}$.}

    $p_{n+1}^{\tilde{w}} = \sum_{i=1}^{n+1} \tilde{w}_i \left[ \mathbb{1}\{v_i > v_{n+1}\} \right]$.

**end if**

**Return:** $p_{n+1}^{\tilde{w}}$

---

Table 6: RegularizedMNISTModel Details

| Component | Details |
|---|---|
| Model Name | RegularizedMNISTModel |
| Purpose | Classification model for MNIST digits (0-9) with robustness to corrupted images |
| Architecture Type | Convolutional Neural Network (CNN) with 3 conv blocks + FC layers |
| Input Shape | (Batch_size, 1, 28, 28) - Grayscale MNIST images |
| Output Shape | (Batch_size, 10) - Logits for 10-class digit classification |
| Total Parameters | ≈600K parameters |
| Layers | First Conv Block: `Conv2d(1→32, 3×3)` → `BatchNorm` → `ReLU` → `MaxPool(2×2)` → `Dropout`
Second Conv Block: `Conv2d(32→64, 3×3)` → `BatchNorm` → `ReLU` → `MaxPool(2×2)` → `Dropout`
Third Conv Block: `Conv2d(64→128, 3×3)` → `BatchNorm` → `ReLU` → `MaxPool(2×2)` → `Dropout`
Fully Connected: `Linear(128×3×3→256)` → `BatchNorm` → `ReLU` → `Dropout`
Output: `Linear(256→10)` |
| Regularization | `Dropout` (rate=0.3), `BatchNormalization` at each layer |
| Activation Functions | `ReLU` throughout the network |
| Loss Function | Cross-entropy |
| Training Epochs | 30 |
| Batch Size | 64 |
| Learning Rate | 0.001 |
| Optimizer | `Adam` |
| Special Features | Extensive regularization for robustness to corrupted images |
| Usage Context | Primary classification model for MNIST digits in conformal prediction framework |

Table 7: ResNet20 Model Details (Standard Format)

| Component | Details |
|---|---|
| Model Name | ResNet20 |
| Purpose | Classification model for CIFAR-10 images |
| Architecture Type | Residual Network (ResNet v1) with basic blocks |
| Input Shape | (Batch_size, 3, 32, 32) - RGB CIFAR-10 images |
| Output Shape | (Batch_size, 10) - Logits for 10-class CIFAR-10 classification |
| Total Parameters | ≈270K parameters |
| Depth | 20 layers (1 initial conv + 18 layers in blocks + 1 final linear) |
| Block Structure | `BasicBlock`: Conv→BN→ReLU→Conv→BN + shortcut connection, followed by `ReLU` |
| Network Architecture | Initial Layer: `Conv2d`(3→16, 3×3) → `BatchNorm` → `ReLU`
Stage 1: 3 `BasicBlocks` (16 channels, stride=1)
Stage 2: 3 `BasicBlocks` (32 channels, stride=2)
Stage 3: 3 `BasicBlocks` (64 channels, stride=2)
Output: `Global AvgPool` → `Linear`(64→10) |
| Regularization | BatchNormalization in each block |
| Activation Functions | ReLU |
| Weight Initialization | Kaiming normal for convolutional layers, constant for batch normalization |
| Loss Function | Cross-entropy |
| Training Epochs | 30 |
| Batch Size | 64 |
| Learning Rate | 0.001 |
| Optimizer | Adam |
| Special Features | Skip connections (residual learning) for better gradient flow |
| Usage Context | Primary classification model for CIFAR-10 in conformal prediction framework |

