# OpenReview forum: "WATCH: Adaptive Monitoring for AI Deployments via Weighted-Conformal Martingales"
_ICML.cc/2025/Conference — ICML 2025 poster_

### Official Review · Reviewer_7BRr · 2025-03-12

**Overall Recommendation:** 4

**Summary:**

The paper proposes an extension of conformal test martingales (CTMs) by the use of weighted conformal p-values rather than standard conformal p-values. The authors argue that such weighted p-values permit testing for more general hypotheses beyond simple (online) exchangeability, and propose a hypothesis that helps disambiguate between covariate and concept/label shifts. A density ratio estimator is employed and the weighted p-values are shown to experimentally adapt to small covariate shifts, whereas stronger covariate and concept shifts are detected. The method is compared to standard CTMs and one type of sequential e-process-based testing on tabular and image data.

**Claims And Evidence:**

The approach is motivated by three goals listed in the introduction and in Figure 1, namely adaptation to small covariate shifts, detection of stronger shifts, and disambiguation on shift origin. From what I can see, all of these goals are addressed to some extent by the method, and are not overclaiming.

I was nonetheless bothered by the claim that  "Our empirical results show that WATCH substantially outperforms SOTA in these areas" which I find to be overly strong, given the actual reported results (e.g. Table 1 and 2), as well as lack of baseline comparisons and some of the issues I had with the experimental designs in general (see below).

**Essential References Not Discussed:**

I've given what I consider relevant references missing or not clearly discussed above.

**Experimental Designs Or Analyses:**

My key issues with this paper are in the experimental section, which reads unclearly and omits all specifics on experimental design, metrics, baselines etc. Instead very similar-looking plots are repeatedly shown (but at times anecdotal, e.g. Fig. 3) to double-down on the difference between WCTM and standard CTM, whereas the comparison to sequential testing ends up allocated to two hard-to-read results tables with no explanations on the considered metrics and their calculation.

Some of the questions I had reading this section where:
- The X-CTM is introduced very late and in a rushed manner, even though it is fundamental to the method. Is the same design used as for WCTM? How would the WCTM perform in benign covariate shifts without the additional information given by the X-CTM, just on its own?
- I didn't follow Eq. 17 on improving the martingale's adaptivity -- is this meant to symbolize a truncation of the past observations?
- How exactly are the density ratio weights computed, and why was this particular estimator chosen? Ablations/comparisons to other options? How can we trust that this estimator is doing a good job, given its crucial role?
- How exactly are the interval widths shrunk in Fig. 2? Is this exclusively due to the obtained weighting, or is there some kind of coverage-based correction?
- Ablations/comparisons on the role of the betting function, which is known to be very important in dictating the growth of the test martingale?
- The classification into benign and severe shifts is somewhat ad-hoc -- a shift is essentially called severe after the fact if the WCTM does not manage to adapt appropriately. E.g. in Fig. 3c the associated shift is a Level-3 corruption on 60% of samples -- who is to say that this is not severe? Rather than rooting those statements in their own method's ability, it seems more principled to root them in some kind of target risk score (e.g. conformal coverage or miscoverage risk reaching some threshold). I see something along those lines in Fig. 2 (left column) where coverage seems satisfied even under benign shift (also for CTMs). Can we see coverage results for sec 4.2 and Table 2?
- Standard CTMs are equipped with the same Type-I error control or false alarm guarantees as the proposed method since they both rely on test martingales. So, I was surprised to see that Table 2 records such high false alarm rates. How can I explain this? Is this because in fact they are designed for the simple exchangeability hypothesis, whereas here you consider the one from Eq. 13, and thus CTMs do not have that guarantee? Then the comparison seems somewhat unfair or not clearly disambiguated. Associatedly, what kind of rejection thresholds are used (these are no-where discussed). Can we see a clear discussion on nominal vs. empirical guarantees and associated empirical false alarm rates for sec 4.2 and 4.3?
- How come that CTMs record higher detection delays than WCTMs, given that they (1) do not try to adapt to shift and (2) are repeatedly shown to be more sensitive and reject faster in the plots? The statements made in L410-413 seem contradictory to that intuition.
- CTMs can also be used without a rejection threshold, merely as a reflection of evidence for occured changepoints that does not disambiguate between severity of shifts. If this was coupled with the tracking of some loss or risk measure to assess severity such that decisions on raising alarms can be taken, what benefit is obtained by WCTM?
- Why are there no baseline comparisons to (1) standard weighted CP methods which are clearly related, e.g. [1,2,3], (2) prior conformal test martingale methods, e.g. [4,5,6], and (3) related sequential testing and changepoint detectors, e.g. [7,8,9], including methods that similarly try to adapt to shifts online? These all seem very relevant, and a meaningful subset thereof would help clarify the benefits of WCTM by highlighting the limitations to each method (e.g. CTM vs. WCP) and how the combination gives new benefits.

[1] Tibshirani, Ryan J., et al. "Conformal prediction under covariate shift." Advances in neural information processing systems 32 (2019).

[2] Barber, Rina Foygel, et al. "Conformal prediction beyond exchangeability." The Annals of Statistics 51.2 (2023): 816-845.

[3] Podkopaev, Aleksandr, and Aaditya Ramdas. "Distribution-free uncertainty quantification for classification under label shift." Uncertainty in artificial intelligence. PMLR, 2021.

[4] Volkhonskiy, Denis, et al. "Inductive conformal martingales for change-point detection." Conformal and Probabilistic Prediction and Applications. PMLR, 2017.

[5] Eliades, Charalambos, and Harris Papadopoulos. "A conformal martingales ensemble approach for addressing concept drift." Conformal and Probabilistic Prediction with Applications. PMLR, 2023.

[6] Eliades, Charalambos, and Harris Papadopoulos. "A betting function for addressing concept drift with conformal martingales." Conformal and Probabilistic Prediction with Applications. PMLR, 2022.

[7] Bar, Yarin, Shalev Shaer, and Yaniv Romano. "Protected test-time adaptation via online entropy matching: A betting approach." Advances in Neural Information Processing Systems 37 (2024): 85467-85499.

[8] Shekhar, Shubhanshu, and Aaditya Ramdas. "Sequential changepoint detection via backward confidence sequences." International Conference on Machine Learning. PMLR, 2023.

[9] Shin, Jaehyeok, Aaditya Ramdas, and Alessandro Rinaldo. "E-detectors: A Nonparametric Framework for Sequential Change Detection." The New England Journal of Statistics in Data Science 2.2 (2023): 229-260.

**Methods And Evaluation Criteria:**

The method (WATCH/WCTM) leverages a single test martingale design adapted from prior work (fixed betting function design, nonconformity scoring function, density ratio estimator, Shiryaev-Roberts supplement, predictor) and is compared against standard CTMs on three tabular datasets with simulated benign shifts (Fig. 2), and against CTMs and a sequential e-process-based testing procedure (not clearly specified which one from that paper) on MNIST and CIFAR-10 with corruptions. Methods are compared in terms of adaptability (for the benign shifts), detection delay, alarm rates, and runtime. Overall the considered criteria seem reasonable, although I don't necessarily think that runtime is a strong metric for this type of online monitoring given they're all relatively fast, and I believe more focus needs to be placed on the disambiguation in terms of (false) alarm rates and maintaining or breakdown of coverage guarantees coinciding with raised alarms.

**Other Comments Or Suggestions:**

- The paper could use a substantial proof-read since there are quite a few typos (e.g. some noted are L88, L186, L171, L301, L302).
- Table 1 is confusing to read since there is a lot of bolding. Perhaps one can omit the blue bolding of the own method.
- The plots in general are too small, and figure legends etc. are only readable if zoomed in. I would suggest a proper overhaul for readability there.
- A clear outline of contributions (e.g. bullet points) in the introduction would be nice to have
- In L140 what is $[0,1]^*$?
- In Eq. 7 the $\forall t$ seems incorrect here, rather its for a given fixed $t$

**Other Strengths And Weaknesses:**

- I liked the motivation in the introduction, the clarity of Fig. 1 to help the reader understand the general problem, and the nice background/exposition in section 2 at the appropriate level of depth. I also enjoyed the connection to the permutation interpretation of weighted p-values in sec 3.1 and 3.2.
- If Eq. 13 ends up being the single hypothesis considered and experimentally examined even though the approach is motivated from a more general hypothesis class perspective $H_0(f)$, then it seems misleading to call sec 3.3. an "Example" unless more are given. Perhaps it should be emphasized that this is the primary testing objective.

**Questions For Authors:**

- Please see questions in the experiments section that I would like to be addressed/clarified.
- Can you clarify the interpretation given in L258-L264 for Eq. 13? To me this seems like a test for covariate shift and not the approximation quality of $\hat{f}^T(X) / f(X)$? Also, does the estimation of only $\hat{f}^T(X)$ still subsume a known $f(X)$, or is it supposed to symbolize an estimated *ratio* rather than estimated shifted covariate distribution?

**Relation To Broader Scientific Literature:**

Even though the general exposition and background in the paper is very nice, I believe the method is not clearly put into context to other related approaches and existing methods. I've provided some references above. One could also think about discussing other ways of combining conformal p-values (beyond within a test martingale framework), e.g. see [1]

[1] Vovk, Vladimir, Bin Wang, and Ruodu Wang. "Admissible ways of merging p-values under arbitrary dependence." The Annals of Statistics 50.1 (2022): 351-375.

**Theoretical Claims:**

The theoretical results are relatively straightforward and rely on key arguments from prior work, with an adaptation to weighted p-values. Thm. 3.1 seems most novel by extending the exchangeability testing argument of conformal p-values to weighted variants. Thm. 3.2 and 3.3. are exclusively reliant on existing arguments (e.g. Ville's Inequality applied to standard CTMs), and simply a direct invocation thereof. In fact, I would suggest to soften the notion of novelty associated with a "Theorem" here -- why not call them Lemmas instead?

On another note, it would be nice to add intuition to the use of Thm. 3.3. Am I to read this as a softening of the high-probability false alarm guarantee given in 3.2 in terms of an expectation term? I am also unsure because it seems to read as a lower bound on the detection delay, but this does not say anything about the "good" performance of the method (i.e. akin to worst-case or upper bound?)

---

> ### Author Rebuttal · Authors · 2025-04-01
>
> Thank you very much for your time, interest, & detailed feedback! We refer to the this anonymous link for supplemental figs: https://sites.google.com/view/authorresponse/home
>
> **Claims and Evidence:**
> - *SOTA statement:* We have removed the quoted statement, & revised it to refer specifically to how our methods compare to the evaluated baselines (regarding adaptation & detection speed).
> - See “Suggested baselines/Relation to Lit” section.
>
> **Methods And Evaluation Criteria:** See “**Spillover from 7BRr**” in response to **oGBa** for:
> - *Specific baseline e-process*
> - *Runtime metric*
> - *Coverage vs alarms*
>
> **Theoretical Claims:**
> - *Thm 3.2 & 3.3:* We are happy to rename “Thm 3.2” to “Proposition 3.2,” and same for 3.3 (we think proposition is more accurate than lemma).
>
> - *Intuition for Thm 3.3:* The guarantee is controlling the average-run length (ARL) *under the null hypothesis*, meaning how long the “scheduled” monitoring procedure can be expected to run without a false alarm before it needs to be reset. What you describe about detection delay would be a guarantee *under an alternative hypothesis;* we do not report such guarantees.
>
> **Experimental Designs Or Analyses:**
>
> - *X-CTM design & ablation of WCTM without X-CTM:* The $X$-CTM uses the same design (ie, same betting function) as the WCTM, but with a nearest-neighbor-distance score. Without a XCTM, a WCTM has 2 main options: (1) no adaptation, which reduces to standard CTM; or, (2) assume that the changepoint occurs at deployment time. See S2.1 at the link for an ablation study on (2).
>
> - *Eq (17):* If every test point is added to the calibration data (as in standard CTMs), then the weights become intractable to compute (Prinster et al. 2024); but, can avoid this by, at some point $t_{ad}$ (given by $X$-CTM), treating the cal data as fixed (no longer adding recent test points to it). (Fixing the cal set results in the WCTM testing a hypothesis that is *conditioned* on the cal set, vs marginal over it; we will add an appendix section to elaborate.)
>
> - *Density-ratio estimator:* The density-ratio weights were estimated via probabilistic classification, as described in Tibshirani et al (2019), Eq (9). Tabular experiments used sklearn’s LogisticRegression; image experiments used a prefit MLP with 2 hidden ReLU layers, fit on 5K pre-change and 5K post-change points. An example ablation study for the effect estimator misspecification is provided in S2.2 at the link provided.
>
> - *Widths shrunk, betting ablations & benign/severe shifts:* **See response #4 to b4ce.**
>
> - *Table 2 questions:* We meant to write “Unnecessary Alarm” to refer to alarms raised when there is a mild/benign shift. The rejection threshold for Table 2 was $10^6$ (as in Fig 3). WCTM unnecessary alarms are evidence that density-ratio estimation is harder for image data. We will add clear discussion of empirical vs nominal rates.
>
> - *CTM higher ADD:* Related to our discussion around Eq (17), once the WCTMs treat the cal data as fixed, they can sometimes reject faster than standard CTMs (the latter adds each test point to the cal set, which confuses the source/target distinction and sacrifices power).
>
> - *Benefit of WCTMs for tracking loss metric:* A loss metric can take the role of the nonconformity score, and WCTMs then would adapt to distribution shifts to maintain statistically valid estimates of how that loss compares to cal data.
>
> - *Suggested baselines/Relation to Lit:* (1) weighted CP methods are not comparable to WCTMs; rather, any WCP method offers a different opportunity to construct a WCTM *on top of it.* Ie, our expts are roughly construct WCTMs on top of WCP methods similar to [1]; in future work it would be valuable to consider WCTMs constructed for continual drift [2] and label shift [3]. We are happy to add comparisons to a subset of (2) and (3) in a camera-ready version, and we will discuss all of these more thoroughly in Related Work. In particular, [4-6] appear to primarily differ from our CTM baseline by betting function and ensembling (which can also be done with WCTMs). [7] is similar to us in the goal of online adaptation, although they focus on adapting the point prediction (while we adapt the weights); we have added more discussion of this. [8] and [9] could be compared to, although the CTM references are likely more closely related.
>
> **Other Weaknesses:** We have updated sec 3.3 subheading to “Main Practical Testing Objective.”
>
> **Other:** We are carefully revising to address typos, to improve clarity and readability. See response #1 to reviewer **b4ce** re contributions outline. $[0,1]^*$ is notation for a vector of any length with entries in [0,1], we will clarify. We meant Eq 7 is the definition for any $t$, we can update if unclear.
>
> **Eq 13 Q:** Should mean a shift to some $\hat{F}_X^T$ *such that* the density-ratio is $\hat{f}_X^T/f_X$. So, $f_X$ does *not* need to be known, only an estimate of the *ratio* is needed. We will update notation here.

---

> > ### Comment · Reviewer_7BRr · 2025-04-05
> >
> > Thank you for the rebuttal and clarifications. The information is scattered everywhere so it was a bit hard to parse. On a high level most of my questions have been answered but fundamentally I am still missing a proper definition of what benign vs. medium vs. extreme constitutes. Even the additional experiments are essentially following the premise that the method's (un)ability to adapt dictates the form of observed shift. Clearly, this categorization is both gradual and problem-specific, so perhaps the authors can instead point out the fact that such a distinction is primarily user-driven, based on the information derived from the martingale's behaviour. Secondly, I am still missing proper explanations and analysis on the false alarm guarantees (not the conformal coverage guarantees but the Type-I sequential testing error), validity of guarantees under different settings and distinction to CTMs, Ramdas' e-process etc. I think these guarantees are a fundamental aspect of the sequential testing paradigm and should clearly be discussed. Similarly, a lack of clarity remains regarding metric formulations, proper explanation of experiment parameters / selected parameter values etc. These should where important be included in the main paper, and else in an experimental details section in the appendix. The paper is overall very dense and difficult to parse beyond the high-level intuition (i.e. in its practical implementation and workings) so I think the paper story should be streamlined to focus on the key aspects and deliver a more focused experimental evaluation with proper discussions, and move additional experiments (of which by now there are many) into the appendix. **I think if the authors properly incorporate these points into the paper it would be substantially strengthened**.
> >
> > Overall, I still like the fundamental ideas of the paper, and I think it combines a few interesting concepts from conformal prediction / sequential testing to address an interesting problem. Albeit novelty is a bit scattered, I think the overall combination and problem setting is of interest to the community. **What is positive to see across all the experiments (incl. rebuttal) is that the approach is fairly consistent in its behaviour**, and even though there is a lack of baselines its results seem promising. So, under the presumption that the authors take these comments to heart I am raising my score.

---

> > > ### Author Response · Authors · 2025-04-09
> > >
> > > Thank you very much for your response! (And, sorry for the info being scattered, due to char limits.) We especially appreciate your further feedback, which we do take to heart, & we agree that it can substantially strengthen our paper--we recognize that incorporating it is in our own interest, for our paper to be as well-received as possible. We plan to revise to add clarifications to each of your questions (including experimental details), streamlining as you describe, acknowledging any limitations (eg, other baselines or settings that could be added), & more.
> > >
> > > **Benign vs medium vs extreme covariate shifts:** First, we will acknowledge in the paper that these distinctions are “gradual and problem-specific” & can be “primarily user-driven.” However, even if partly subjective, we do not think they are arbitrary--in particular, the distinction we tried to make so far is that benign shifts are those where the “safety” of CP coverage is maintained nontrivially. Ie, the coverage-validity intuition for “benign” shifts corresponds to the WCTM’s null hypothesis (& harmful shifts violate it) as follows:
> > >
> > > - *Benign:* The martingale’s null hypothesis is that the WCP $p$-values are IID uniform (App. C.2); this null implies that coverage is satisfied (exactly) for all $\alpha\in [0,1]$ (ie, the martingale’s null $\implies$ intuitive definition of “benign” regarding coverage validity).
> > >
> > > - *Harmful:* (Contrapositive of the above.) If coverage is *not* satisfied (exactly) for some $\alpha\in [0,1]$, then the $p_t$ are *not* IID Uniform[0,1] (ie, violation of coverage validity $\implies$ violation of martingale’s null, thus possibility for detection).
> > >
> > >     - Note: This can be due to under- or over-coverage (we further penalize trivial overcoverage, where $\hat{C}(X_{n+1})=\mathcal{Y}$, as in the link’s S1.2).
> > >
> > > - *Medium:* A shift may initially be “harmful” as described above, due to density-ratio estimator having insufficient data, but later become “benign” once it has enough data.
> > >
> > > We will note that further study of robustness (ie, when is density-ratio estimator “good enough”) & power (ie, how quickly can harmful shifts be detected) are important directions for future work. Additionally, regarding the difficulty/subjectivity in interpreting martingale values, we note that this is not unique to WCTMs; eg, Jeffreys’s “rule of thumb” is cited in related work including Vovk (2021) "Testing Randomness Online" (pg 601) for interpreting the strength of evidence.
> > >
> > >
> > > **Explanations & analysis on the sequential false alarm guarantees:** We agree that it is important for the anytime-valid (3.2) & scheduled/ARL (3.3) sequential false alarm guarantees to be thoroughly explicated, analyzed, & empirically evaluated; we will further revise so that these points are even clearer in the paper.
> > >
> > > - WCTMs achieve the same strength of guarantees as CTMs, but under different null hypotheses: our (3.2) has the same form as “strong validity” in Vovk (2021) & our (3.3) takes the form of his prop 4.2 in that paper; the difference is that *WCTMs achieve these guarantees under null hypotheses different than exchangeability* (we practically focus on the covariate shift null). The anytime-valid guarantee (3.2) controls the probability of raising a false alarm (despite null being true) at *any* time in an infinite run; meanwhile, the average-run length (ARL) guarantee (3.3) lower bounds the expected “expiration date” of the method until it needs to be reset, which is why it is called “scheduled” or “multistage.”
> > >
> > > - Re evaluations, as we state at the end of Sec 4.1, the blue in 3rd column of Fig 2 empirically validates Thm (3.1); 4th column validates (3.2); 5th column validates (3.3).
> > >
> > > - In Sec 4.3, we compare to the Ramdas e-process baseline by standardizing the (W)CTM & e-processes’ false alarm rates under their respective nulls at $P(alarm) \leq 0.01=1/c$ for anytime-valid methods & $ARL \geq 20,000=c$ for scheduled methods (see “Specific baseline e-process” in response to reviewer oGBa).
> > >
> > > - (Time permitting) We hope to add additional baselines mentioned by the reviewer, and/or an additional distribution shift setting.
> > >
> > >
> > > **Experimental details:** We have added full experimental details for the image-data experiments to the provided link’s S3 (Experiment Details), & we will add these tables to the paper appendix (with key information in the main paper); we will also double-check that all synthetic-data & tabular-data experimental details as well (or add any details that were not included before). We will ensure that all metrics are clearly defined.
> > >
> > >
> > > **Streamlining paper:** We appreciate, agree with, & will do our best to incorporate the reviewer’s suggestions on how the paper can be streamlined to be more accessible & clear, such as with a more focused experimental evaluation in the main paper.
> > >
> > >
> > > **Summary & thank you:** Again, thank you for your comments! We assure that we will work faithfully to incorporate your feedback & address your comments.

---

### Official Review · Reviewer_oGBa · 2025-03-14

**Overall Recommendation:** 3

**Summary:**

This paper proposes WATCH, a novel method that is able to check machine learning models after deployment to identify if their input data changes unexpectedly. WATCH uses a new approach, Weighted Conformal Test Martingales (WCTMs), to detect these changes. WATCH can ignore small data changes that do not affect the model much and only raises warnings when big changes occur. The authors conduct experiments on real datasets and show that WATCH performs better than previous methods at correctly identifying important data shifts without many false warnings.

**Claims And Evidence:**

- The claim on the contribution of using weighted conformal p-Values could be discussed more clearly. After reading section 3.1, i feel i am not clear on whether weighted conformal p-values are novel in this paper. I would like the authors to indicate clearly on how this is related to prior works such as [1, 2].

- Is the notion of the concept shift defined before? I couldn't find any discussion on this. I think this is important, given that this paper shows benefit most on this type of shift. So i wonder if it is new or it is considered an important task in this area before. I would love the authors to provide more evidence in this aspect to support the claim of the strong results.


[1] Conformal validity guarantees exist for any data distribution (and how to find them). 2024.
[2] Conformal prediction under covariate shift. 2019.

**Essential References Not Discussed:**

I don't have concerns on this aspect.

**Experimental Designs Or Analyses:**

I find no issue in the soundness or validity of the experimental design or analysis. The evaluation mostly follow prior setups in this area.

**Methods And Evaluation Criteria:**

- In the experiment section, the evaluated models are mostly simple neural networks, which seems rather limited (e.g., MNIST and CIFAR-10). I wonder if the evaluation can be conducted on large models and datasets, for example, ViT on ImageNet.

- In the paper, the authors mostly consider covariate shift or concept shift, I wonder if there are other types of distribution shifts that could exist. I would like to see more discussion on this aspect.

**Other Comments Or Suggestions:**

None

**Other Strengths And Weaknesses:**

- I am wondering if the authors could discuss more on how this result is practical, like in a real-world setting, how can it be used in large-scale machine learning systems. This could help reader better understand the impact of the paper.

- If the shift is detected, then what can we do to address the problem? Are there ways to leverage the proposed detection method to improve the model dynamically at deployment time? If so, how would you do it?

**Questions For Authors:**

None.

**Relation To Broader Scientific Literature:**

I think the idea of detecting changes in the test data distribution could have many practical applications. The first is to monitor the deployment setting in real time. Also, this could help model developers adapt the model better to address potential catastrophical failures.

**Theoretical Claims:**

I check the proof of theorem 3.1, 3.2 and 3.3, which is correct. But i would suggest to highlight the idea of the proof in the main text, rather than deferring them entirely to the Appendix.

---

> ### Author Rebuttal · Authors · 2025-04-01
>
> Thank you for your time and feedback! Please find our responses to your questions/comments roughly in order below. We refer to the following anonymous link for supplemental figures and algorithms: https://sites.google.com/view/authorresponse/home
>
> **Claims and Evidence:**
> - *Novelty of weighted-conformal $p$-values:* The general weighted-conformal $p$-values presented in Section 3 are novel in this paper. References [1] and [2] have a similar setup based on a weighted empirical distribution of nonconformity scores, but they only consider taking quantiles on the weighted distribution to compute prediction sets, they do not define, introduce, or mention weighted-conformal $p$-values nor any equivalent quantity, and they do not discuss how weighted conformal ideas can be used for hypothesis testing or monitoring (the focus of our paper).
>
> - *References for concept shift:* Concept shift--a shift in $Y|X$, the label distribution conditional on the input--is a common term and topic of study in the ML robustness literature. The earliest refs we could find discussing the idea of concept shift are the following--we have added these to the paper: (a) Webb, G. I., & Ting, K. M. (2005). On the application of ROC analysis to predict classification performance under varying class distributions. Machine learning, 58, 25-32; (b) Fawcett, T., & Flach, P. A. (2005). A response to Webb and Ting’s on the application of ROC analysis to predict classification performance under varying class distributions. Machine Learning, 58, 33-38. While prior CTM methods are also capable of detecting concept shift, our main novel contribution regarding concept shift is to enable valid continual monitoring for concept shift *even when covariate shift is also possible,* i.e., by enabling diagnosis between the two, and demonstrating fast detection empirically. At the link we have provided above, we have added additional ablation experiments on the simple synthetic data to illustrate how this behavior.
>
> **Methods And Evaluation Criteria:**
> - *Eval on larger models and datasets:* The theoretical guarantees of our methods make no assumptions on the ML model architecture or on the modality/complexity/size of the dataset, so yes in theory we could demonstrate similar performance with ViT on ImageNet. Our paper focuses on laying a theoretical and algorithmic foundation for WCTM-based monitoring methods, while opening the door to future evaluation on frontier and foundation models.
>
> - *Other types of shifts:* Yes, there are other types of shifts that can be studied. One example is “label shift,” which is a shift in the marginal $Y$ distribution (but not in $Y|X$); another example is slow distribution drift over time (rather than at a single changepoint). We have added discussion of WCTMs for these settings to our Conclusion.
>
> **Other Strengths And Weaknesses:**
> - *Practical use in large-scale ML system:* In large-scale ML deployments, the proposed WATCH framework could be used to monitor the performance of the system to detect harmful behavior patterns (e.g., increases in LLM hallucinations, jailbreaking attacks, toxic prompts/generated text), while also adapting online to more benign shifts (e.g., seasonality, user trends, etc). For instance, WCTMs could be run to monitor the distribution of LLM outputs while serving as a “safety filter” to provide probabilistic guarantees on the safety of the output shown to users. The massive computational cost of training large models paired with their wide-reaching effects makes the task of monitoring crucial, to quickly catch unsafe behavior while also minimizing costly retraining.
>
> - *Improving model dynamically at deployment time:* WATCH implements one approach to dynamically improving the model at deployment time, that is by performing online density-ratio estimation to maintain prediction safety (coverage) and informativeness (widths). Root-cause analysis can also inform other approaches to improving performance at deployment time, eg, via distributionally robust training (to be robust to detected shifts).
>
> **Spillover from Reviewer 7BRr (char lim)--Methods & Evaluation Criteria:**
> - *Specific baseline e-process:* We compared to the betting-based $e$-process in Podkopaev & Ramdas (2021) (from Waudby-Smith & Ramdas (2024)) since those methods performed best and to standardize. We compared WCTMs to sequential testing variant with standardized anytime-false alarm rate of 0.01; for SR-WCTMs we compared vs changepoint detection variant with common average-run-length (under the null) of 20,000.
>
> - *Runtime metric:* Runtime can be surprisingly nontrivial for monitoring long time-horizons. Ie, the changepoint baseline in Podkopaev & Ramdas (2021) has $O(t^2)$ complexity (due to starting a new sequential test at each time $t$); our comparable SR-WCTM is only $O(t)$.
>
> - *Coverage vs alarms:* In our Appendix, we plan to add coverage & width plots for all expts in our paper; some are available already at the link.

---

> > ### Comment · Reviewer_oGBa · 2025-04-01
> >
> > I have read the rebuttal and my concern was mostly addressed. It would be better to show some results on larger models and datasets. Overall the paper is technically novel and therefore I maintain my score.

---

> > > ### Author Response · Authors · 2025-04-09
> > >
> > > Thank you for your response! In particular, thank you for confirming that your concerns have mostly been addressed and for recognizing the technical novelty of our contributions! We agree that it would be promising and valuable to also demonstrate results on larger models and datasets (eg, ViT on ImageNet) as an expansion on our current empirical results (which currently include neural networks evaluated on synthetic data, real tabular data, and real image data [MNIST & CIFAR-10]). Demonstrations on larger models and datasets is an important practical direction that our paper opens the door to--especially given that our methods require no formal assumptions/restrictions on the model, or on the size, modality, or dimensionality of the dataset--but we leave this direction open for future work.

---

### Official Review · Reviewer_b4ce · 2025-03-15

**Overall Recommendation:** 2

**Summary:**

This works proposes a weighted generalization of conformal test martingales (WCTMs), for online change point detection, which can continuously adapt to benign shifts without raising unnecessary alarms and quickly detect harmful shifts.

**Claims And Evidence:**

partially, see "Other Comments Or Suggestions" below, mainly the distinction between mild covariate shifts and extreme covariate shifts is not clear.

**Essential References Not Discussed:**

N/A, and I am not fully familiar with all literature in this area.

**Experimental Designs Or Analyses:**

yes

**Methods And Evaluation Criteria:**

yes

**Other Comments Or Suggestions:**

The mild covariate shifts and extreme covariate shifts are a bit vague in this paper, without clear distinction in terms of, for example, the covariate shift magnitude. Only with a clearer definition of benign shifts and harmful shifts, users can adjust the boundary between benign and harmful and tailor the detection algorithm accordingly. Due to this reason, the overall detection algorithm seems vague to me, and I would appreciate it if the author(s) could provide a detailed algorithm (including the choice of f functions) for performing the detection online, and justify how the algorithm treats benign and harmful shifts separately.

The proposed method seems to depend a lot on Vovk 2021 and Vovk 2022. And it should be discussed whether the weights added is the only difference or maybe there are more innovations as compared with the previous literature.

**Other Strengths And Weaknesses:**

Strengths: a new weighted version of CTM that can adapt to benign changes and detect harmful changes quickly.

Weakness: the writing can be improved, and more mathematical concepts and a clearer definition of benign and harmful changes will make the problem setup easier to understand.

**Questions For Authors:**

see above.

**Relation To Broader Scientific Literature:**

it is related to the broad community of online learning and distributional change detection, and is mostly related with Vovk 2021 and Vovk 2022 which studies the non-weighted version of conformal test martingale.

**Theoretical Claims:**

partially, the theorems seems to be correct to the best of my knowledge.

---

> ### Author Rebuttal · Authors · 2025-04-01
>
> Thank you for your time and feedback! We refer to the following anonymous link for supplemental figures and algorithms: https://sites.google.com/view/authorresponse/home
>
> (1) **Clarifying writing, especially novelty of contributions relative to Vovk et al:** Regarding your comment on how “writing can be improved,” we are actively revising our paper for clarity. For example, re your comment on clarifying “innovations as compared with the previous literature,” we have revised the last part of our Introduction section to the following (this references a slightly revised Fig 1, available at the provided link):
>
> Recent advances in anytime-valid inference (Ramdas et al., 2023) and especially conformal test martingales (CTMs) (Volkhonskiy et al., 2017; Vovk, 2021) offer promising tools for AI monitoring with sequential, nonparametric guarantees. However, existing CTM monitoring methods (e.g., Vovk et al., (2021)) all rely on some form of exchangeability (e.g., IID) assumption in their null hypotheses---informally, meaning that the data distribution is the same across time or data batches---and as a result, standard CTMs can raise unnecessary alarms even when a shift is mild or benign (e.g., Figure 1a). Meanwhile, existing comparable monitoring methods for directly tracking the risk of a deployed AI (e.g., Podkopaev and Ramdas (2021b)) tend to be less efficient than CTMs regarding their computational complexity, data usage, and/or speed in detecting harmful shifts (Sec. 4.3).
>
> Our paper's contributions can be summarized as follows:
>
> - Our main theoretical contribution is to propose weighted-conformal test martingales (WCTMs), constructed from weighted-conformal $p$-values, which generalize their standard conformal precursors. WCTMs lay a theoretical foundation for online testing of a broad class of null hypotheses beyond exchangeability, including shifts that one aims to model and adapt to.
>
> - For practical applications, we propose WATCH: Weighted Adaptive Testing for Changepoint Hypotheses, a framework for AI monitoring using WCTMs. WATCH continuously adapts to mild or benign covariate shifts (e.g., Figure 1a) to maintain end-user safety and utility (and avoid unnecessary alarms), while quickly detecting harmful shifts (e.g., Figure 1b \& c) and enabling root-cause analysis.
>
>
> (2) **Additional Explanatory Ablation Experiments on Covariate Shift Magnitude:** To address your comment that our distinction between mild and extreme covariate shifts are somewhat “vague,” we have added additional ablation experiments to illustrate how WATCH performs with different magnitudes of covariate shift--please see Supplement S1.1 at the provided link. Ie, this ablation experiment empirically clarifies that covariate shifts can indeed be viewed as a spectrum from benign to harmful rather than as a binary category. For the monitoring purpose that our paper focuses on, however, we consider “benign” shifts to be those that can be safely ignored without sacrificing system safety (i.e., coverage) or system informativeness/utility (i.e., prediction set sharpness/interval width); our WCTM methods can thus be understood as performing a flexible (nonparametric) statistical test for whether a given shift is benign or harmful to the system, in this sense, with sequential false-discovery guarantees given by our reported theoretical results.
>
>
> (3) **Added Pseudocode, Highlighting How Algorithm Treats Benign vs. Harmful Shifts:** We are adding comprehensive pseudocode for all proposed algorithms as a new appendix section (Appendix E). For now, we have provided the annotated pseudocode for the specific sub-module of our methods that responds slightly differently to mild versus extreme shifts in $X$--please see Supplement S.1.2 for this pseudocode at the provided link. Specifically, the pseudocode highlights one way that our methods penalize (and thereby more quickly detect) covariate shifts that are so extreme (i.e., out-of-support or very little support) that the corresponding weighted conformal interval becomes noninformative (i.e., predicting the whole label space, $\hat{C}(X_{n+1})=\mathcal{Y}$ ).
>
>
> (4) **Spillover responses for Reviewer 7BRr (due to char lim) -- Methods And Evaluation Criteria:**
>
> - *Widths shrunk in Fig 2:* Due to the density-ratio estimator appropriately placing more weight on cal points with smaller scores--no further correction.
>
> - *Ablations on betting function:* In S2.3 at the provided link, we compare the Composite Jumper betting function to the Simple Jumper betting function used in baselines (eg, Vovk et al).
>
> - *Classification into benign and severe shifts:* See responses #2 & #3 to reviewer b4ce. Regarding a “target risk score,” note that (W)CTMs test for IID uniformity on the $p$-values; IID uniform $p$-values implies valid coverage at any significance level, which is more powerful than for a threshold. So, we argue our def is at least as principled. See sec 4.2 at the link provided for coverage results.

---

### Decision · Program_Chairs · 2025-05-01

**Decision:**

Accept (poster)

**Comment:**

The paper proposes a novel approach called WATCH, which is based on a weighted generalization of conformal test martingales, which automatically adapts to mild data shifts while detecting substantial shifts and raising alarms. Compared to conformal test martingales with uniform weight, the novel, generalized version can test a broader range of null hypotheses beyond exchangeability. Specifically, authors show that in the case where the estimated density ratio function is used as the weight, the method can detect either an *extreme* covariate shift or concept shift, while under moderate covariate shift, the method is able to adequately adapt its conformal prediction interval. The method ensures control over the false alarm rate. Theoretical claims are supported by experiments.

Overall, reviewers agree on the novelty of the proposed approach. However, the paper could be strengthened by clarifying:
-the exact distinction between ‘benign’, ‘medium’, and ‘harmful’ covariate shifts (based on the WCTM values)
-novelty of the proposed ‘weighted-conformal-p-values’
-details on the experiments (including metrics, experiment parameters, interpretation of the results)